# LLM-FE: Automated Feature Engineering for Tabular Data with LLMs as Evolutionary Optimizers

**Nikhil Abhyankar**     *nikhilsa@vt.edu*
**Parshin Shojaee**     *parshinshojaee@vt.edu*
**Chandan K. Reddy**     *reddy@cs.vt.edu*
*Department of Computer Science, Virginia Tech*

**Reviewed on OpenReview:** *https://openreview.net/forum?id=qvI35hkpOO*

## Abstract

Automated feature engineering plays a critical role in improving predictive model performance for tabular learning tasks. Traditional automated feature engineering methods are limited by their reliance on pre-defined transformations within fixed, manually designed search spaces, often neglecting domain knowledge. Recent advances using Large Language Models (LLMs) have enabled the integration of domain knowledge into the feature engineering process. However, existing LLM-based approaches use direct prompting or rely solely on validation scores for feature selection, failing to leverage insights from prior feature discovery experiments or establish meaningful reasoning between feature generation and data-driven performance. To address these challenges, we propose LLM-FE, a novel framework that combines evolutionary search with the domain knowledge and reasoning capabilities of LLMs to automatically discover effective features for tabular learning tasks. LLM-FE formulates feature engineering as a program search problem, where LLMs propose new feature transformation programs iteratively, and data-driven feedback guides the search process. Our results demonstrate that LLM-FE consistently outperforms state-of-the-art baselines, showcasing generalizability across diverse models, tasks, and datasets.

The code is available at:  https://github.com/nikhilsab/LLMFE

## 1 Introduction

Feature engineering, the process of transforming raw data into meaningful features for machine learning models, is crucial for improving predictive performance, particularly when working with tabular data (Domingos, 2012). In many tabular prediction tasks, well-designed features have been shown to significantly enhance the performance of tree-based models, often outperforming deep learning models that rely on learned representations (Grinsztajn et al., 2022). However, data-centric tasks such as feature engineering are one of the most time-consuming and resource-intensive processes in the tabular learning workflow (Anaconda, 2020; Hollmann et al., 2024), as they require experts and data scientists to explore many possible combinations in the vast combinatorial space of feature transformations. Classical feature engineering methods (Kanter & Veeramachaneni, 2015; Khurana et al., 2016; 2018; Horn et al., 2020; Zhang et al., 2023) construct extensive search spaces of feature processing operations, relying on various search and optimization techniques to identify the most effective features. However, these search spaces are mostly constrained by predefined, manually designed transformations and often fail to incorporate domain knowledge (Zhang et al., 2023). Domain knowledge can serve as an invaluable prior for identifying these transformations, leading to reduced complexity and more interpretable and effective features (Hollmann et al., 2024).

Recently, Large Language Models (LLMs) have emerged as a powerful solution to this challenge, offering access to extensive embedded domain knowledge that can be leveraged for feature engineering. While recent approaches have demonstrated promising results in incorporating this knowledge into automated feature discovery, current LLM-based methods (Hollmann et al., 2024; Han et al., 2024) predominantly rely on direct prompting or validation scores to guide feature generation. These approaches do not leverage insights from prior feature-discovery experiments, thereby falling short of establishing a meaningful link between feature generation and data-driven performance.

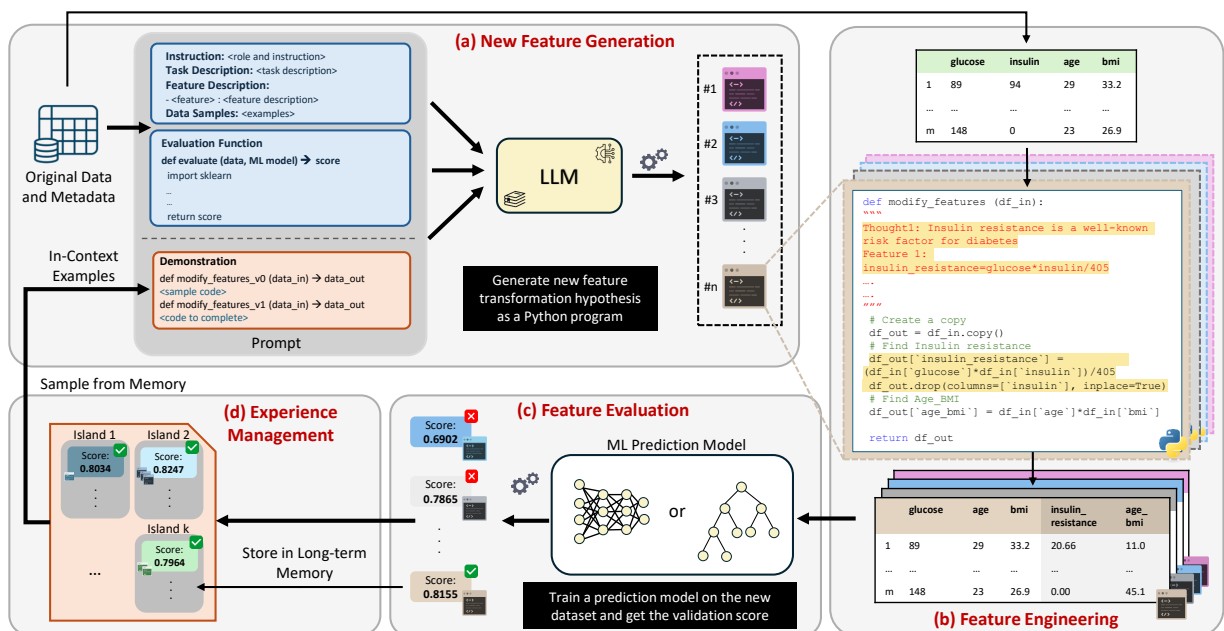

Figure 1: **Overview of the LLM-FE Framework.** For a given dataset, LLM-FE follows these steps: (a) **New Feature Generation**, where an LLM generates feature transformation hypotheses as programs for a given tabular dataset; (b) **Feature Engineering**, where the feature transformation program is applied to the underlying dataset, resulting in a modified dataset; (c) **Feature Evaluation**, where the modified dataset with the new features is evaluated using a prediction model; (d) **Experience Management**, which maintains a buffer of high-scoring programs that act as in-context samples for LLM's iterative refinement prompt. The features generated by LLM-FE are interpretable, using the LLM's domain knowledge.

To address these limitations, we propose **LLM-FE**, *a novel framework integrating the capabilities of LLMs with table prediction models and evolutionary search to facilitate effective feature optimization.* As shown in Figure 1, LLM-FE follows an iterative process to generate and evaluate the hypothesis of the feature transformation, using the performance of the tabular prediction model as a reward to enhance the generation of effective features. Starting from an initial feature transformation program, LLM-FE leverages the LLMs' embedded domain knowledge by incorporating task-specific details, feature descriptions, and a subset of data samples to generate new feature discovery programs (Figure 1(a)). At each iteration, the LLM acts as a knowledge-guided evolutionary optimizer, mutating examples of previously successful feature transformation programs to generate new, effective features (Meyerson et al., 2024). The newly proposed features are then integrated with the original dataset to yield an augmented dataset (Figure 1(b)). The prediction model's performance is evaluated on a held-out validation set derived from the augmented dataset (Figure 1(c)), provides data-driven feedback that, combined with a dynamic memory of previously explored feature transformation programs (Figure 1(d)), guides the LLM to iteratively refine its feature generation.

Table 1 compares LLM-FE to several state-of-the-art classical and LLM-based feature engineering methods. Traditional methods lack adaptability and deeper contextual understanding, while LLM-based methods generate simple features (Küken et al., 2024) or use feedback to iteratively refine only a single rule. In contrast, LLM-FE supports all four aspects by leveraging

Table 1: Comparison of existing feature engineering methods.

| Method | Domain Knowledge | Feedback Driven | Complex Features | Multi-Feature Refinement |
|---|---|---|---|---|
| AutoFeat (Horn et al., 2020) | ✗ | ✗ | ✓ | ✗ |
| OpenFE (Zhang et al., 2023) | ✗ | ✗ | ✓ | ✗ |
| FeatLLM (Han et al., 2024) | ✓ | ✗ | ✗ | ✗ |
| CAAFE (Hollmann et al., 2024) | ✓ | ✓ | ✗ | ✗ |
| OCTree (Nam et al., 2024) | ✓ | ✓ | ✗ | ✗ |
| **LLM-FE** | ✓ | ✓ | ✓ | ✓ |

LLM-based domain knowledge and feedback-driven optimization to generalize well across tabular prediction tasks. We evaluate LLM-FE with `Llama-3.1-8B-Instruct` (Dubey et al., 2024) and `GPT-3.5-Turbo` (OpenAI, 2023) backbones on classification and regression tasks across diverse tabular datasets. LLM-FE consistently outperforms the state-of-the-art feature engineering methods, identifying contextually relevant features that improve downstream performance. In particular, we observe improvements with tabular models like `XGBoost` (Chen & Guestrin, 2016), `TabPFN` (Hollmann et al., 2023), and `MLP` (Gorishniy et al., 2021), highlighting the importance of evolutionary search in achieving effective results. The major contributions are:

• We introduce LLM-FE, a novel framework that casts feature engineering as an LLM-guided evolutionary optimization problem, integrating domain knowledge, data-driven evaluation, and long-term memory for iterative refinement.

• Our experimental results demonstrate the effectiveness of LLM-FE, showcasing its ability to outperform state-of-the-art baselines, demonstrating generalizability across different predictors and LLM backbones.

• Through a comprehensive ablation study, we highlight the critical role of domain knowledge, evolutionary search, data-driven feedback, and data samples in guiding the LLM to efficiently explore the feature space and discover impactful features more effectively.

## 2    Related Works

**Feature Engineering.**    Feature engineering involves creating meaningful features from raw data to improve predictive performance (Hollmann et al., 2024). The growing complexity of datasets has driven the automation of feature engineering to reduce manual effort and optimize feature discovery. Traditional automated feature engineering methods include tree-based exploration, transformation enumeration, and learning-based methods (Khurana et al., 2016; Kanter & Veeramachaneni, 2015; Nargesian et al., 2017; Zhang et al., 2023). These traditional approaches often fail to leverage domain knowledge for feature discovery, making LLMs well-suited for such tabular prediction tasks due to their prior contextual domain understanding.

**LLMs and Optimization.**    Advances in LLMs have shown that they can adapt to novel tasks via prompt engineering and in-context learning without retraining (Brown et al., 2020; Wei et al., 2022). Yet, their outputs can be inconsistent or factually incorrect (Madaan et al., 2024; Zhu et al., 2023), motivating research into mechanisms that refine or stabilize generations. A growing body of work has explored coupling LLMs with evaluators in iterative or evolutionary frameworks, where feedback, mutation, and crossover guide the search for solutions (Lehman et al., 2023; Wu et al., 2024; Meyerson et al., 2024). This paradigm has yielded progress in prompt optimization (Yang et al., 2024b; Guo et al., 2024), neural architecture search (Zheng et al., 2023; Chen et al., 2023), mathematical heuristic discovery (Romera-Paredes et al., 2024), and symbolic regression (Shojaee et al., 2025). Building on this trajectory, our LLM-FE framework operationalizes LLMs as evolutionary optimizers, combining their rich prior knowledge with systematic, data-driven refinement to discover compact and high-performing features.

**LLMs for Tabular Learning.**    The application of LLMs to structured data has typically relied on converting tables into textual representations (Dinh et al., 2022; Hegselmann et al., 2023; Wang et al., 2023), or tailoring tokenization and pre-training strategies for tabular robustness (Yan et al., 2024). For tabular prediction specifically, LLMs have been employed in fine-tuning and few-shot in-context paradigms (Hegselmann et al., 2023; Nam et al., 2023), as well as in direct feature engineering. For example, FeatLLM (Han et al., 2024) generates binary rules, while CAAFE (Hollmann et al., 2024) exploits task descriptions to generate contextual features, and OCTree (Nam et al., 2024) iteratively improves features through decision tree reasoning. However, these approaches often rely on incremental refinement of a single candidate. In contrast, LLM-FE maintains a diverse pool of promising programs and employs evolutionary search to efficiently traverse the feature space, leveraging mutation and crossover to uncover interpretable and data-driven transformations. This design enables the discovery of features that are not only predictive but also interpretable by humans, bridging the gap between domain-informed reasoning and optimization. Appendix A further illustrates the qualitative differences between LLM-FE and the baseline methods.

## 3    LLM-FE Approach

### 3.1    Problem Formulation

A tabular dataset $\mathcal{D}$ comprises $N$ rows (or instances), each characterized by $d$ columns (or features). Each data instance $x_i$ is a $d$-dimensional feature vector with feature names denoted by $C = \{c_j\}_{j=1}^{d}$. The dataset is accompanied by metadata $\mathcal{M}$ that includes feature descriptions and task-specific information. For supervised learning tasks, each instance $x_i$ is associated with a corresponding label $y_i$, where $y_i \in \{0, 1, ..., K\}$ for classification tasks with $K$ classes, and $y_i \in \mathbb{R}$ for regression tasks. Given a labeled tabular dataset $\mathcal{D} = (x_i, y_i)_{i=1}^{N}$ and prediction model $f$ to map from the input feature space $\mathcal{X}$ to its corresponding label space $\mathcal{Y}$, the feature engineering objective is to determine an optimal feature transformation program $\mathcal{T}$, which maps the original feature space $\mathcal{X}$ to an augmented representation $\mathcal{T}(\mathcal{X})$ that improves predictive

performance when used to train the downstream model. Thus, the feature engineering task can be defined as:

$$\max_{\mathcal{T}} \mathcal{E}(f^*(\mathcal{T}(\mathcal{X}_{\text{val}})), \mathcal{Y}_{\text{val}}), \tag{1}$$

subject to

$$f^* \in \arg\min_{f} \mathcal{L}_f(f(\mathcal{T}(\mathcal{X}_{\text{tr}})), \mathcal{Y}_{\text{tr}}), \tag{2}$$

where $(\mathcal{X}_{tr}, \mathcal{Y}_{tr})$ and $(\mathcal{X}_{val}, \mathcal{Y}_{val})$ are the sub-training set and validation set, respectively, that are derived from the training data $(\mathcal{X}_{train}, \mathcal{Y}_{train})$. The feature transformation program $\mathcal{T}$ is generated by the LLM $\pi_\theta$ conditioned on a prompt $p$. Candidate programs are therefore sampled as $\mathcal{T} \sim \pi_\theta(p)$ where $p$ is constructed from dataset metadata $\mathcal{M}$, data samples $\mathcal{X}$, and high-scoring programs from prior iterations (detailed in Section 3.2). The predictive model $f^*$ is then trained on the transformed training data $\mathcal{T}(\mathcal{X}_{train})$ to minimize loss. The bilevel objective in Eqs. (1)–(2) is intractable to optimize directly over the combinatorial space of feature transformation programs. LLM-FE therefore approximates it via evolutionary search at each iteration (see Algorithm 1), where $\pi_\theta$ proposes candidate programs $\{\mathcal{T}_j\}$ conditioned on $p$, each is evaluated by training $f^*$ on $\mathcal{T}_j(\mathcal{X}_{tr})$ and scoring on $\mathcal{T}_j(\mathcal{X}_{val})$, serving as a fitness signal guiding iterative refinement.

## 3.2 Feature Generation

Figure 1(a) illustrates the feature generation step that uses an LLM to create multiple new feature transformation programs, leveraging the model's prior knowledge, reasoning, and in-context learning abilities to effectively explore the feature space.

### 3.2.1 Input Prompt

To facilitate the creation of effective and contextually relevant feature discovery programs, we develop a structured prompting methodology. The prompt is designed to provide comprehensive data-specific information, an initial feature transformation program for the evolution starting point, an evaluation function, and a well-defined output format (see Appendix B.1 for more details). Our input prompts $p$ are composed of the following key elements:

**Instruction.** The LLM is assigned the task of finding the most relevant features to help solve the given task. The task emphasizes using the LLM's prior knowledge of the dataset's domain to generate features. The LLM is explicitly instructed to generate novel features and provide clear step-by-step reasoning for their relevance to the prediction task. Moreover, since LLMs tend to generate simple features, we specifically instruct the LLM to generate complex features.

**Dataset Specification.** After providing the instructions, we present the LLM with the dataset-specific information from the metadata $\mathcal{M}$. This information encompasses a detailed description of the intended downstream task, along with the feature names $C$ and their corresponding descriptions. In addition, we provide a limited number of representative samples from the tabular dataset. To improve the effective interpretation of the data, we adopt the serialization approach used in previous works (Dinh et al., 2022; Hegselmann et al., 2023; Han et al., 2024). We serialize the data samples as follows:

$$\text{Serialize}(x_i, y_i, C) = \text{'If } c_1 \text{ is } x_i^1, ..., c_d \text{ is } x_i^d. \text{ Then Result is } y_i\text{'}. \tag{3}$$

By providing dataset-specific details, we guide the language model to focus on the most contextually pertinent features that directly support the dataset and task objective.

**Evaluation Function.** The evaluation function, incorporated into the prompt, guides the language model to generate feature-transformation programs that align with the performance objectives. These programs augment the original dataset with new features, which are evaluated based on the performance of a prediction model trained on the augmented data. The model's evaluation score on the augmented validation set serves as an indicator of feature quality. By including the evaluation function in the prompt, the LLM generates programs that are inherently aligned with the desired performance criteria.

**In-Context Demonstration.** Specifically, we sample the $k$ highest-performing demonstrations from previous iterations, enabling the LLM to build on successful outputs. The iterative interaction between the LLM's generative outputs and the evaluator's feedback, informed by these examples, facilitates a systematic refinement process. With each iteration, the LLM progressively improves its outputs by leveraging patterns and insights identified in previous successful demonstrations.

### 3.2.2 Feature Sampling

At each iteration $t$, we construct the prompt $p_t$ by sampling the previous iteration as input to the LLM $\pi_\theta$, resulting in the output $\mathcal{T}_1, \ldots, \mathcal{T}_b = \pi_\theta(p_t)$ representing a set of $b$ sampled programs. To promote diversity and maintain a balance between exploration (creativity) and exploitation (prior knowledge), we employ stochastic temperature-based sampling. Each of the sampled feature transforms ($\mathcal{T}_i$) is executed before evaluation to discard error-prone programs. This ensures that only valid, executable feature transformation programs are considered in the optimization pipeline. In addition, to ensure computational efficiency, a maximum execution time threshold is enforced, and any program that exceeds it is discarded.

### 3.3 Data-Driven Evaluation

As illustrated in Figure 1(b), we use the generated features to augment the original dataset with the newly derived features. Similar to (Hollmann et al., 2024; Nam et al., 2024), our feature evaluation process comprises two stages: (i) model training on the augmented dataset, and (ii) performance assessment for feature quality (Figure 1(c)). We fit a tabular predictive model $f^*$, to the transformed training set $\mathcal{T}(\mathcal{X}_{\mathrm{tr}})$, by minimizing the loss $\mathcal{L}_f$ as shown in Eq. (1). Subsequently, we evaluated the LLM-generated feature transformations $\mathcal{T}$ by calculating the model's performance on the augmented validation set $\mathcal{T}(\mathcal{X}_{\mathrm{val}})$ (see Eqs.1 and 2). As explained in Section 3.1, the objective is to find optimal features that maximize the performance $\mathcal{E}$, i.e., accuracy for classification and error metrics for regression.

### 3.4 Experience Management

To promote diverse feature discovery and avoid stagnation in local optima, LLM-FE employs evolutionary multi-population experience management (Figure 1(d)) to store feature discovery programs in a dedicated database. Then, it uses samples from this database to construct in-context examples for LLM, facilitating the generation of novel features. This step consists of two components: (i) multi-population memory to maintain a long-term memory buffer, and (ii) sampling from this memory buffer to construct in-context example demonstrations. After evaluating the feature transforms in iteration $t$, we store the pair of feature transforms and score $(\mathcal{T}, s)$ in the population buffer $\mathcal{P}_t$ to iteratively refine the search process. To effectively evolve a population of programs, we adopt a multi-population model inspired by the 'island' model employed by Cranmer (2023); Romera-Paredes et al. (2024); Shojaee et al. (2025). The program population is divided into $m$ independent islands, each having access to the full original feature set but evolving separately and initialized

---

**Algorithm 1** LLM-FE

**Require:** Dataset $\mathcal{D}$, Metadata $\mathcal{M}$, Iterations $T$, Model $f$, LLM $\pi_\theta$, Metric $\mathcal{E}$
1: $\mathcal{P}_0 \leftarrow \texttt{BufferInit}()$
2: $\mathcal{T}^*, s^* \leftarrow \text{null}, -\infty$
3: $p \leftarrow \texttt{UpdatePrompt}(\mathcal{D}, \mathcal{M})$
4: **for** $t = 1$ to $T$ **do**
5: $\quad p_t \leftarrow p + \mathcal{P}_{t-1}.\texttt{topk}()$
6: $\quad \{\mathcal{T}_j\}_{j=1}^b \leftarrow \pi_\theta(p_t)$
7: $\quad$ **for** $j = 1$ to $b$ **do**
8: $\quad\quad s_j \leftarrow \texttt{FeatureScore}(f, \mathcal{T}_j, \mathcal{D}, \mathcal{E})$
9: $\quad\quad$ **if** $s_j > s^*$ **then**
10: $\quad\quad\quad \mathcal{T}^*, s^* \leftarrow \mathcal{T}_j, s_j$
11: $\quad\quad$ **end if**
12: $\quad\quad \mathcal{P}_t \leftarrow \texttt{UpdateBuffer}(\mathcal{P}_{t-1}, \mathcal{T}_j, s_j)$
13: $\quad$ **end for**
14: **end for**
15: **return** $\mathcal{T}^*, s^*$

---

with a copy of the user's initial example (see Figure 9(d)). This enables parallel exploration of the feature space, mitigating the risk of suboptimal solutions. At each iteration $t$, we select one of the $m$ islands and sample programs from the memory buffer to update the prompt with new in-context examples.

The newly generated feature samples $b$ are evaluated, and if their scores $s_j$ exceed the current best score, the feature score pair $(\mathcal{T}_j, s_j)$ is added to the same island from which the in-context examples were sampled. To preserve diversity and ensure that programs with different performance characteristics remain in the buffer, we cluster programs within each island based on their program signature, defined by their validation performance score $s$. In particular, feature transformation programs that produce identical validation scores are consolidated into the same cluster. Following Romera-Paredes et al. (2024); Shojaee et al. (2025), we first sample from one of the $m$ available islands, followed by sampling the $k$ programs from the selected island to create $k$-shot in-context examples for the LLM (see Appendix B.1). Cluster selection is performed using Boltzmann sampling (De La Maza & Tidor, 1992), which assigns a higher probability to clusters with higher scores, using a score-based probability for choosing a cluster $i$: $P_i = \frac{exp(s_i/\tau_c)}{\sum_i exp(s_i/\tau_c)}$, where $s_i$ denotes the mean score of the $i$-th cluster and $\tau_c$ is the temperature parameter. The temperature $\tau_c$ controls the exploration–exploitation trade-off: lower values concentrate probability mass on the highest-scoring

cluster (exploitation), while higher values spread mass more evenly across clusters. The sampled feature transformation programs from the memory buffer are then included in the prompt as in-context examples to guide the LLM toward generating more effective feature transformations. Detailed descriptions of the memory management strategy, clustering procedure, and sampling mechanism are provided in the Appendix B.1. Algorithm 1 presents the pseudocode of LLM-FE. We begin by initializing a memory buffer `BufferInit` with an initial population that includes a simple feature transform. This initialization serves as the starting point for the evolutionary search for feature transformation programs to be evolved in the subsequent steps. At each iteration $t$, the function `topk` is used to sample $k$ in-context examples from the population of the previous iteration $\mathcal{P}_{t-1}$ to update the prompt. Subsequently, we prompt the LLM using this updated prompt to sample $b$ new programs. The sampled programs are then evaluated using `FeatureScore`, which represents the Data-Driven Evaluation (Section 3.3). After $T$ iterations, the best-scoring program $\mathcal{T}^*$ from $\mathcal{P}_t$ and its score $s^*$ are returned as the optimal solution found for the problem. LLM-FE employs an iterative search to enhance programs, harnessing the LLM's capabilities. Learning from the evolving pool of experiences in its buffer, the LLM steers the search toward effective solutions. Thus, unlike classical evolutionary algorithms that apply explicit mutation or crossover operators, LLM-FE performs implicitly through prompt-conditioned generation. Previously successful programs, included as in-context examples, guide the LLM to produce variants of individual programs and combinations of features from programs.

## 4 Experimental Setup

We evaluated LLM-FE on a range of tabular datasets, encompassing classification and regression tasks. Our experimental analysis included quantitative comparisons with baselines and detailed ablation studies. Specifically, we assessed our approach using three known tabular predictive models with distinct architectures: (1) `XGBoost`, a tree-based model (Chen & Guestrin, 2016), (2) `MLP`, a neural model (Gorishniy et al., 2021), and (3) `TabPFN` (Hollmann et al., 2023), a transformer-based foundation model (Vaswani et al., 2017). The results highlight LLM-FE's ability to generate effective features that consistently improve the performance of various prediction models across datasets.

### 4.1 Baselines

We evaluated LLM-FE against state-of-the-art feature engineering approaches, including OpenFE (Zhang et al., 2023) and AutoFeat (Horn et al., 2020), as well as LLM-based methods CAAFE (Hollmann et al., 2024), FeatLLM (Han et al., 2024), and OCTree (Nam et al., 2024). We used `XGBoost` as the default tabular data prediction model in comparison with baselines and employed `GPT-3.5-Turbo` as the default LLM backbone for all LLM-based methods (Tables 2 and 3). To ensure a fair comparison, all LLM-based approaches operate under a fixed budget of 20 LLM samples with no early stopping or convergence criterion; the final reported result corresponds to the best-scoring program discovered within this budget. Appendix B.2 contains additional implementation details.

### 4.2 Datasets

We followed Hollmann et al. (2024) to select datasets from previous feature engineering work, such as Han et al. (2024); Hollmann et al. (2024); Zhang et al. (2023), that include descriptive feature information. Our analysis contains 19 classification and 10 regression datasets, each containing mixed categorical and numerical features. We also include 8 large-scale, high-dimensional classification datasets to ensure comprehensive evaluation. These datasets were sourced from established machine learning repositories, including OpenML (Vanschoren et al., 2014; Feurer et al., 2021), UCI Machine Learning Repository (Asuncion et al., 2007), and Kaggle. Furthermore, we conduct experiments on five classification datasets from Hollmann et al. (2024) and Bordt et al. (2024), released after the September 2021 GPT training cutoff date. Each dataset is accompanied by metadata, including a natural-language description of the prediction task and descriptive feature names. We partitioned each dataset into train and test sets using an 80-20 split. Following Hollmann et al. (2024), we evaluated all methods over five iterations, each time using a distinct random seed and train-test splits. For more details, check Appendix C.

### 4.3 LLM-FE Configuration

In our experiments, we used `GPT-3.5-Turbo` and `Llama-3.1-8B-Instruct` as backbone LLMs, with a sampling temperature of $t = 0.8$ and $m = 3$ islands. At each iteration, the LLM generated $b = 3$ feature transformation programs per prompt in Python. To ensure consistency with baselines, LLM-FE was also

configured to use 20 LLM samples per experiment. Finally, we sampled the top $m$ (where $m$ denotes the number of islands) feature discovery programs based on their respective validation scores. More implementation details are provided in Appendix B.1.

Table 2: **Performance of XGBoost on Classification Datasets using various Feature Engineering (FE) Methods**, evaluated using accuracy (higher values indicate better performance). We report the mean values and standard deviation across five splits. ✗: denotes execution time of greater than 12 hours (for classical FE methods) or failure due to execution errors (for LLM-based FE methods). **bold:** indicates the best performance. underline: indicates the second-best performance. 'n': indicates the number of samples; 'p': indicates the number of features.

| Dataset | n | p | Base | Classical FE Methods | | LLM-based FE Methods | | | LLM-FE |
| --- | --- | --- | --- | --- | --- | --- | --- | --- | --- |
| | | | | AutoFeat | OpenFE | CAAFE | FeatLLM | OCTree | |
| adult | 48.8k | 14 | $0.873_{\pm 0.002}$ | ✗ | $0.873_{\pm 0.002}$ | $0.872_{\pm 0.002}$ | $0.842_{\pm 0.003}$ | $0.870_{\pm 0.002}$ | **$0.874_{\pm 0.003}$** |
| arrhythmia | 452 | 279 | $0.657_{\pm 0.019}$ | ✗ | ✗ | ✗ | ✗ | ✗ | **$0.659_{\pm 0.018}$** |
| balance-scale | 625 | 4 | $0.856_{\pm 0.020}$ | $0.925_{\pm 0.036}$ | $0.986_{\pm 0.009}$ | $0.966_{\pm 0.029}$ | $0.800_{\pm 0.037}$ | $0.882_{\pm 0.022}$ | **$0.990_{\pm 0.013}$** |
| bank-marketing | 45.2k | 16 | $0.906_{\pm 0.003}$ | ✗ | $0.906_{\pm 0.002}$ | **$0.907_{\pm 0.002}$** | **$0.907_{\pm 0.002}$** | $0.900_{\pm 0.002}$ | **$0.907_{\pm 0.002}$** |
| breast-w | 699 | 9 | $0.956_{\pm 0.012}$ | $0.956_{\pm 0.019}$ | $0.956_{\pm 0.014}$ | $0.960_{\pm 0.009}$ | $0.967_{\pm 0.014}$ | $0.969_{\pm 0.009}$ | **$0.970_{\pm 0.009}$** |
| blood-transfusion | 748 | 4 | $0.742_{\pm 0.012}$ | $0.738_{\pm 0.014}$ | $0.747_{\pm 0.025}$ | $0.749_{\pm 0.017}$ | **$0.771_{\pm 0.016}$** | $0.755_{\pm 0.026}$ | $0.751_{\pm 0.036}$ |
| car | 1728 | 6 | $0.995_{\pm 0.003}$ | $0.998_{\pm 0.003}$ | $0.998_{\pm 0.003}$ | **$0.999_{\pm 0.001}$** | $0.808_{\pm 0.037}$ | $0.995_{\pm 0.004}$ | **$0.999_{\pm 0.001}$** |
| cdc diabetes | 253k | 21 | $0.849_{\pm 0.001}$ | ✗ | $0.849_{\pm 0.001}$ | $0.849_{\pm 0.001}$ | $0.849_{\pm 0.001}$ | $0.849_{\pm 0.001}$ | $0.849_{\pm 0.001}$ |
| cmc | 1473 | 9 | $0.528_{\pm 0.012}$ | $0.505_{\pm 0.015}$ | $0.517_{\pm 0.007}$ | $0.524_{\pm 0.016}$ | $0.479_{\pm 0.015}$ | $0.525_{\pm 0.027}$ | **$0.531_{\pm 0.019}$** |
| communities | 1.9k | 103 | $0.706_{\pm 0.016}$ | ✗ | $0.704_{\pm 0.009}$ | $0.707_{\pm 0.013}$ | $0.593_{\pm 0.012}$ | $0.708_{\pm 0.016}$ | **$0.711_{\pm 0.012}$** |
| covtype | 581k | 54 | $0.870_{\pm 0.001}$ | ✗ | **$0.885_{\pm 0.007}$** | $0.872_{\pm 0.003}$ | $0.554_{\pm 0.001}$ | $0.832_{\pm 0.002}$ | $0.882_{\pm 0.003}$ |
| credit-g | 1000 | 20 | $0.751_{\pm 0.019}$ | $0.757_{\pm 0.017}$ | $0.758_{\pm 0.017}$ | $0.751_{\pm 0.020}$ | $0.707_{\pm 0.034}$ | $0.753_{\pm 0.021}$ | **$0.766_{\pm 0.015}$** |
| eucalyptus | 736 | 19 | $0.655_{\pm 0.024}$ | $0.664_{\pm 0.028}$ | $0.663_{\pm 0.033}$ | **$0.679_{\pm 0.024}$** | ✗ | $0.658_{\pm 0.041}$ | $0.668_{\pm 0.027}$ |
| heart | 918 | 11 | $0.858_{\pm 0.013}$ | $0.857_{\pm 0.021}$ | $0.854_{\pm 0.020}$ | $0.849_{\pm 0.023}$ | $0.865_{\pm 0.030}$ | $0.852_{\pm 0.022}$ | **$0.866_{\pm 0.021}$** |
| jungle_chess | 44.8k | 6 | $0.869_{\pm 0.001}$ | ✗ | $0.900_{\pm 0.004}$ | $0.901_{\pm 0.038}$ | $0.577_{\pm 0.002}$ | $0.869_{\pm 0.002}$ | **$0.969_{\pm 0.004}$** |
| myocardial | 1.7k | 111 | $0.784_{\pm 0.023}$ | ✗ | $0.787_{\pm 0.026}$ | **$0.789_{\pm 0.023}$** | $0.778_{\pm 0.023}$ | $0.787_{\pm 0.031}$ | **$0.789_{\pm 0.023}$** |
| pc1 | 1109 | 21 | $0.931_{\pm 0.004}$ | $0.931_{\pm 0.009}$ | $0.931_{\pm 0.004}$ | $0.929_{\pm 0.005}$ | $0.933_{\pm 0.007}$ | $0.934_{\pm 0.007}$ | **$0.935_{\pm 0.006}$** |
| tic-tac-toe | 958 | 9 | $0.998_{\pm 0.002}$ | **$1.000_{\pm 0.000}$** | $0.994_{\pm 0.006}$ | $0.996_{\pm 0.003}$ | $0.653_{\pm 0.037}$ | $0.997_{\pm 0.003}$ | $0.998_{\pm 0.005}$ |
| vehicle | 846 | 18 | $0.754_{\pm 0.016}$ | **$0.788_{\pm 0.018}$** | $0.785_{\pm 0.008}$ | $0.771_{\pm 0.019}$ | $0.744_{\pm 0.035}$ | $0.753_{\pm 0.036}$ | $0.769_{\pm 0.013}$ |
| **Mean Rank** | | | 3.95 | 5.11 | 3.63 | 3.47 | 5.11 | 4.05 | **1.42** |

Table 3: **Performance of XGBoost on Regression Datasets using various Feature Engineering (FE) Methods,** evaluated using RMSE (lower values indicate better performance). We report the mean values and standard deviation across five splits. **bold:** indicates the best performance. underline: indicates the second-best performance. 'n': indicates the number of samples; 'p': indicates the number of features.

| Dataset | n | p | Base | Classical FE Methods | | LLM-based FE Methods | | LLM-FE |
| --- | --- | --- | --- | --- | --- | --- | --- | --- |
| | | | | AutoFeat | OpenFE | Base LLM | OCTree | |
| airfoil_self_noise $[10^0]$ | 1503 | 6 | $1.572_{\pm 0.084}$ | $1.531_{\pm 0.118}$ | $1.631_{\pm 0.111}$ | $1.507_{\pm 0.150}$ | $1.572_{\pm 0.079}$ | **$1.451_{\pm 0.059}$** |
| bike $[10^1]$ | 17389 | 11 | $4.094_{\pm 0.096}$ | $4.222_{\pm 0.123}$ | $4.089_{\pm 0.140}$ | $4.149_{\pm 0.101}$ | $4.094_{\pm 0.096}$ | **$3.985_{\pm 0.084}$** |
| cpu_small $[10^0]$ | 8192 | 10 | $2.857_{\pm 0.223}$ | $2.896_{\pm 0.197}$ | $2.822_{\pm 0.190}$ | $2.798_{\pm 0.226}$ | $2.832_{\pm 0.192}$ | **$2.733_{\pm 0.249}$** |
| crab $[10^0]$ | 3893 | 8 | $2.325_{\pm 0.094}$ | $2.266_{\pm 0.078}$ | $2.221_{\pm 0.010}$ | $2.309_{\pm 0.135}$ | $2.280_{\pm 0.087}$ | **$2.211_{\pm 0.124}$** |
| diamond $[10^2]$ | 53940 | 9 | $5.479_{\pm 0.063}$ | $5.521_{\pm 0.143}$ | $5.384_{\pm 0.084}$ | $5.422_{\pm 0.075}$ | $5.479_{\pm 0.063}$ | **$5.356_{\pm 0.134}$** |
| forest-fires $[10^1]$ | 517 | 13 | $0.163_{\pm 0.009}$ | $0.163_{\pm 0.010}$ | $0.161_{\pm 0.013}$ | $0.165_{\pm 0.018}$ | $0.162_{\pm 0.007}$ | **$0.156_{\pm 0.008}$** |
| housing $[10^4]$ | 20640 | 9 | $4.845_{\pm 0.191}$ | $4.776_{\pm 0.271}$ | $4.628_{\pm 0.105}$ | $4.961_{\pm 0.457}$ | $4.845_{\pm 0.191}$ | **$4.525_{\pm 0.260}$** |
| insurance $[10^3]$ | 1338 | 7 | $5.269_{\pm 0.260}$ | $5.098_{\pm 0.323}$ | $5.085_{\pm 0.286}$ | $5.112_{\pm 0.362}$ | **$4.969_{\pm 0.331}$** | $5.069_{\pm 0.392}$ |
| plasma_retinol $[10^2]$ | 315 | 13 | $2.352_{\pm 0.196}$ | $2.478_{\pm 0.217}$ | $2.363_{\pm 0.195}$ | $2.384_{\pm 0.200}$ | $2.362_{\pm 0.204}$ | **$2.278_{\pm 0.248}$** |
| wine $[10^0]$ | 4898 | 10 | $0.639_{\pm 0.006}$ | $0.633_{\pm 0.007}$ | $0.631_{\pm 0.009}$ | $0.639_{\pm 0.009}$ | $0.639_{\pm 0.006}$ | **$0.612_{\pm 0.007}$** |
| **Mean Rank** | | | 4.55 | 4.45 | 2.80 | 4.40 | 3.70 | **1.10** |

## 4.4 Results and Discussion

In Table 2, we compare LLM-FE against various feature engineering baselines across 19 classification datasets. The results demonstrate that LLM-FE consistently improves predictive performance relative to the base model (using raw data). LLM-FE also achieves the lowest mean rank (best performance) at a lower computational cost (see Section 5.1), demonstrating better effectiveness in enhancing feature discovery than other leading baselines. To further evaluate the effectiveness of LLM-FE, we perform experiments on 10 regression datasets using the same evaluation settings employed for the classification datasets. Due to the lack of regression data implementations in the available codebases for LLM-based baselines (CAAFE and FeatLLM), we restrict our comparison in Table 3 to non-LLM methods (OpenFE & AutoFeat), base LLM, and OCTree, which have been previously validated on regression tasks. The results indicate that LLM-FE outperforms all baseline methods, achieving the lowest mean rank and consistently improving across all datasets. We provide additional analyses in Appendix D, including Wilcoxon signed-rank tests, analysis of the effect of hyperparameter optimization on LLM-FE, and evaluations with alternative predictive models such as CatBoost and Logistic Regression.

We further study the transferability and generalizability of discovered features across different LLM backbones, showing that LLM-FE remains robust and effective under varied modeling and architectures.

### 4.5 Ablation Study

We perform an ablation study on the classification datasets (<10,000 samples) listed in Table 2 to assess the contribution of each component in LLM-FE. Figure 2 illustrates the impact of individual components on overall performance, using `XGBoost` and `GPT-3.5-Turbo`. We report the accuracy aggregated and normalized over all the datasets. In the **'w/o Domain Knowledge'** setting, dataset and task-specific details are removed from the prompt and feature names are anonymized with generic placeholders such as $C_1$, $C_2, \ldots, C_n$. In this way, we remove any semantic meaning that could provide contextual insights about the problem. Without domain knowledge, the performance significantly drops to 0.626, underscoring its critical role in generating meaningful features. The **'w/o Evolutionary Refinement'** setting also leads to the greatest decline in performance (0.587), emphasizing the importance of iterative data-driven feedback in addition to domain knowledge for refining feature transforms. Lastly, the results show that **'w/o Data Examples'** variant leads to only a slight performance drop, as

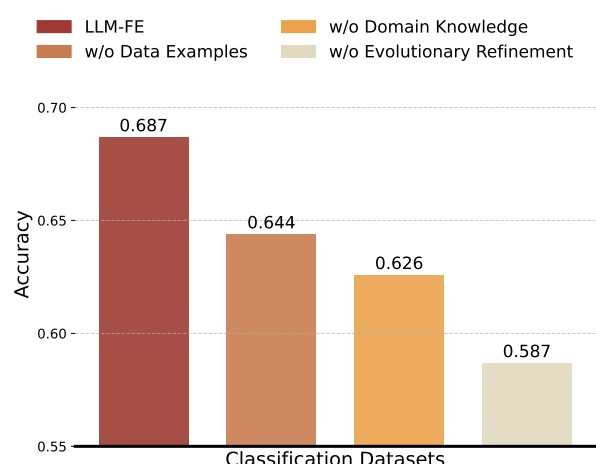

Figure 2: **Aggregated ablation study results across classification datasets**, showcasing the impact of individual components on LLM-FE's performance: (a) Data Examples, (b) Domain Knowledge, and (c) Evolutionary Refinement. Values are normalized with respect to the base LLM-FE model to facilitate fair comparison.

LLMs might struggle to comprehend the nuances and patterns within the data samples. LLM-FE benefits significantly from each component, leading to an improvement.

## 5 Analysis

### 5.1 Efficiency Analysis

Evaluating feature quality through repeated model training and validation is a fundamental component of automated feature engineering pipelines, used for both classical and LLM-based methods. All runtime measurements were obtained on a system with 4 NVIDIA RTX8000 GPUs. For LLM-based methods, runtime measures the full pipeline from the start of the first iteration to the completion of the last iteration, including LLM API latency, feature program execution, and model training. For classical baselines, runtime includes the complete feature generation, selection, and evaluation pipeline. While LLM-FE maintains multiple evolutionary islands, this design does not introduce additional computational overhead in practice. Furthermore, LLM-FE samples multiple outputs per LLM call, reducing the number of API calls. Our Pareto analysis (Figure 3) on the larger datasets from Section 4.4 shows that LLM-FE consistently lies on the Pareto frontier, achieving higher predictive performance with nearly

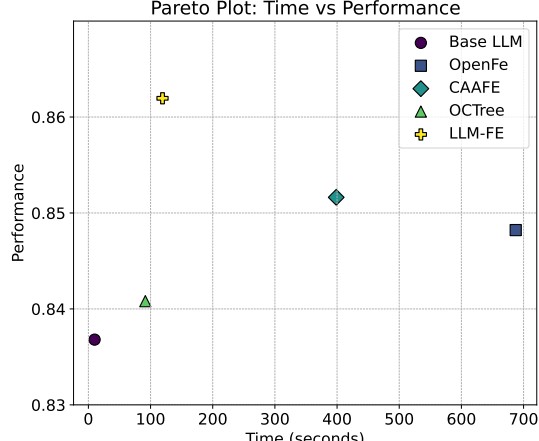

Figure 3: **Pareto Plot:** comparing trade-off between performance (accuracy) vs time (in seconds) for LLM-FE and other FE baselines.

the same runtime as OCTree and substantially less runtime than CAAFE. Competing approaches either require substantially more runtime (CAAFE, OpenFE) or fail to reach comparable accuracy (OCTree). Although the base model remains the cheapest computationally, it does so at a steep performance deficit. Overall, these results demonstrate that LLM-FE offers the best efficiency–performance trade-off among all evaluated methods. It achieves state-of-the-art predictive performance on large, complex tabular datasets, with no additional overhead introduced.

## 5.2 Bias Mitigation

LLMs also exhibit a pronounced bias toward a narrow set of simple mathematical operators such as addition, subtraction, and absolute value when asked to generate feature transformations (Küken et al., 2024). These biases arise from the pretraining corpora, where simple patterns dominate and thus become default strategies. As a result, naive LLM-based feature engineering pipelines tend to produce repetitive, low-complexity transformations that fail to exploit the richer compositional space of meaningful tabular operations. As illustrated in Figure 4, CAAFE also tends to favor simple transformations with `multiply` and `divide` operations covering up to 75% of the total operators. Despite this inherent bias, LLM-FE regularly discovers and retains more sophisticated feature transformations through evolutionary refinement. Operators such as `groupbythenmean`, `groupbythenmin`, `groupbythenmax`, `residual`, and `sigmoid` emerge far more frequently under our evolutionary framework than in direct LLM generation. These higher-level operations capture aggregation structure, class-conditional variation, and nonlinear relationships that simple arithmetic cannot express. This outcome highlights an important point: while LLMs alone are biased toward oversimplified transformations, our evolutionary search mechanism actively counteracts this tendency by (1) promoting diversity, (2) evaluating transformations through empirical performance, and (3) iteratively refining candidate features. Consequently, LLM-FE not only reduces the risk of memorization but also mitigates operator-selection bias, enabling the discovery of expressive, domain-relevant features that would rarely surface through naive prompting alone.

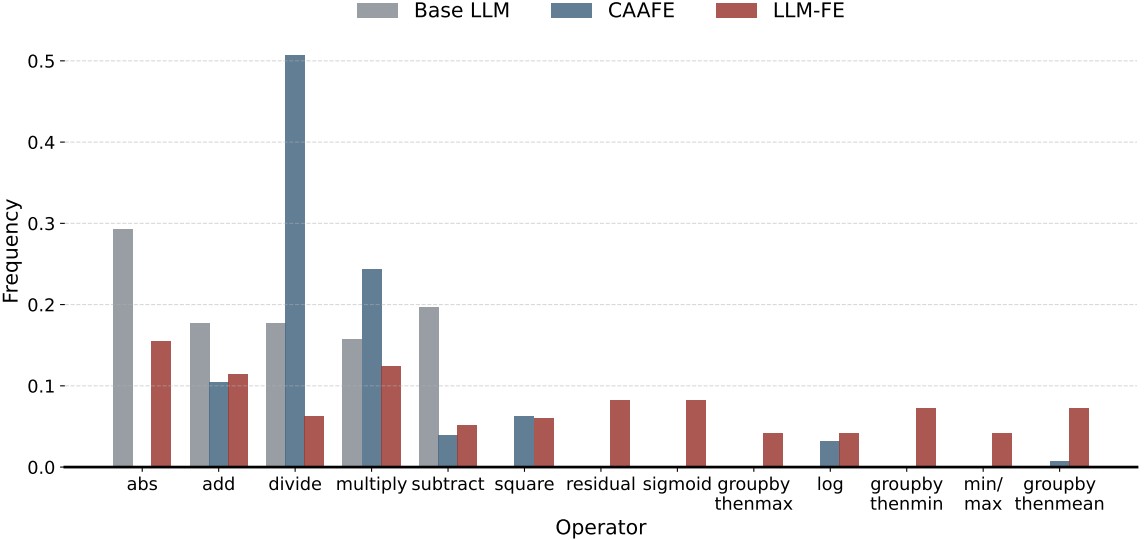

Figure 4: **Frequency of Feature Engineering Operators.** Comparison of operator usage between LLM-FE and simple LLM baselines, highlighting the ability of evolutionary refinement to extract more complex transformations.

## 5.3 Impact of Domain Knowledge and Evolutionary Refinement

Figure 7 illustrates the qualitative benefits of incorporating domain knowledge into feature engineering. In this example, two approaches are contrasted: one without domain knowledge (Figure 7(a)), and LLM-FE guided by domain-specific insights through an LLM-based feature engineering (Figure 7(b)). The domain-agnostic variant creates arbitrary transformations, such as combining features `C1` and `C3` using a square root of their product and dropping feature `C2` without clear justification. In contrast, LLM-FE leverages its embedded knowledge to derive interpretable and clinically meaningful features. Figure 5 presents a quantitative comparison of model performance on the same dataset, showing that LLM-FE with domain knowledge achieves the highest accuracy, outperforming both the base model and LLM-FE without domain knowledge. Figure 6 illustrates the validation accuracy trajectory of LLM-FE with and without evolutionary refinement across 20 iterations. The variant without refinement shows early improvement but quickly plateaus, indicating convergence to a local optimum. In contrast, LLM-FE continues to improve across iterations, achieving higher accuracy overall. This comparison highlights the effectiveness of evolutionary refinement in enhancing performance by enabling the model to escape local optima and optimize more effectively. Further analyses are provided in Appendix E.

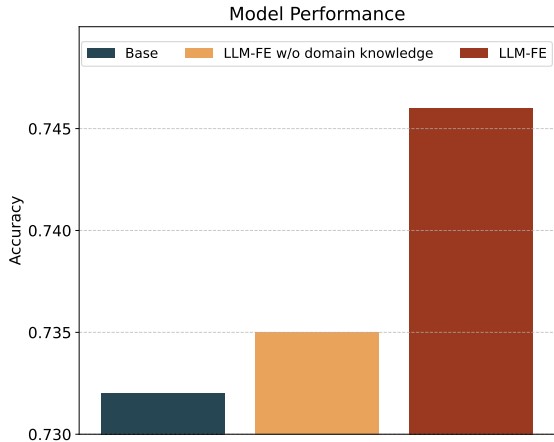

Figure 5: **Quantitative impact of domain knowledge on model accuracy.** Using domain knowledge boosts performance compared to both the base model and LLM-FE without domain knowledge.

Figure 6: **Performance Trajectory Analysis** for LLM-FE *w/o* evolutionary refinement and LLM-FE. LLM-FE demonstrates a better trajectory, highlighting the advantage of evolutionary refinement.

(a) Feature Engineering without domain knowledge

```
def modify_features(df_input) -> pd.DataFrame:
    """
    Introducing a new feature 'C10' as the square root of the
    product of 'C1' and 'C3' to capture a non-linear relationship
    between these variables.
    Additionally, dropping less informative feature 'C2'.
    """
    df_output = df_input.copy()

    df_output['C10'] = np.sqrt(df_output['C1'] * df_output['C3'])
    df_output.drop('C2', axis=1, inplace=True)

    return df_output
```

(b) LLM-FE

```
def modify_features(df_input) -> pd.DataFrame:
    """
    Thought 1: Insulin levels in conjunction with Glucose levels can provide
        insights into the metabolic state.
    Feature 1: insulin_glucose_ratio |
        insulin_glucose_ratio = Insulin / Glucose
    Thought 2: BMI can be an indicator of potential diabetes risk, especially
        when combined with age.
    Feature 2: bmi_age_ratio | bmi_age_ratio = BMI / Age
    """
    df_output = df_input.copy()

    # Calculate Insulin divided by Glucose
    df_output['insulin_glucose_ratio'] = df_output['Insulin'] /
                                                    df_output['Glucose']

    # Calculate BMI divided by Age
    df_output['bmi_age_ratio'] = df_output['BMI'] / df_output['Age']

    return df_output
```

Figure 7: **Qualitative Analysis on Impact of Domain Knowledge.** illustrating how LLM-FE (b) utilizes domain knowledge to create meaningful features with descriptions , in contrast to feature engineering without domain insights (a) leading to uninterpretable outputs.

## 5.4 Memorization in Feature Engineering

Recent work demonstrates that LLMs can unintentionally memorize data under certain conditions (Carlini et al., 2021; Bordt et al., 2024), raising concerns about whether improvements result from genuine LLM reasoning or merely from recalling training examples. To probe this issue, we evaluate XGBoost with and without LLM-FE using `GPT-3.5-Turbo` on datasets introduced by Bordt et al. (2024), which are explicitly constructed to detect memorization and are confirmed to be absent from model pretraining. We further include datasets from Hollmann et al. (2024), released after the September 2021 GPT training cutoff and provided on Kaggle with hidden splits, making pretraining exposure highly unlikely. As shown in Table 4, LLM-FE delivers modest but consistent performance gains across all datasets. This is a notable contrast to naive LLM feature generators, which may inadvertently overfit or hallucinate domain relationships. Instead of relying solely on the raw outputs of the LLM, LLM-FE iteratively selects, mutates, and evaluates candidate features based on downstream model performance. This process acts as a filter, systematically suppressing memorization-driven artifacts and promoting features that generalize across repeated evaluations. While memorization remains an important risk in LLM-driven tabular workflows, our results indicate that evolutionary refinement provides an effective safeguard, underscoring the need for future benchmarks that isolate and stress-test these behaviors.

Table 4: Comparison of XGBoost with and without LLM-FE on five classification datasets.

| Dataset | Base | LLM-FE |
|---|---|---|
| kidney-stones | **0.761 ± 0.024** | **0.761 ± 0.027** |
| health-insurance | 0.756 ± 0.001 | **0.759 ± 0.001** |
| pharyngitis | 0.655 ± 0.008 | **0.660 ± 0.023** |
| fico | 0.715 ± 0.006 | **0.719 ± 0.009** |
| acs-income | 0.807 ± 0.002 | **0.809 ± 0.003** |

## 5.5 Generalizability Analysis

To evaluate the generalizability of LLM-FE, we conduct a systematic assessment of its performance across multiple tabular prediction models and diverse LLM backbones. In particular, we consider two representative LLMs, `Llama-3.1-8B-Instruct` and `GPT-3.5-Turbo`, and pair them with three widely-used tabular prediction models: `XGBoost` (Chen & Guestrin, 2016), a strong tree-based baseline for structured data; a Multilayer Perceptron (`MLP`), which provides a simple yet competitive deep-learning architecture for tabular inputs (Gorishniy et al., 2021); and `TabPFN` (Hollmann et al., 2023), a recent transformer-based foundation model tailored specifically to tabular learning. As summarized in Table 5, our results consistently show that LLM-FE identifies informative and task-relevant features that improve the downstream performance of all three prediction models under both LLM backbones. Moreover, we observe that feature sets generated by LLM-FE reliably outperform their non-feature-engineering counterparts, suggesting that the method provides robust benefits across model classes, backbone choices, and tasks, underscoring its broadly applicable feature-engineering framework. Appendix D.2 presents additional experiments with additional LLMs and predictive models, further demonstrating the generalizability of our method.

Table 5: **Performance improvement by LLM-FE using different prediction models and LLM backbones.** We report the aggregated values for accuracy on classification tasks and normalized root mean square error on regression tasks. All results represent the mean and standard deviation computed across five splits. **bold:** indicates the best performance. TabPFN[*] : indicates evaluations using only 10,000 samples due to its limited processing capacity.

| Method | LLM | Classification ↑ | Regression ↓ |
|---|---|---|---|
| XGBoost | | | |
| Base | – | $0.820 \pm 0.020$ | $0.324 \pm 0.016$ |
| LLM-FE | Llama 3.1-8B | $0.832 \pm 0.021$ | $0.310 \pm 0.022$ |
| | GPT-3.5 Turbo | $\mathbf{0.840} \pm \mathbf{0.022}$ | $\mathbf{0.306} \pm \mathbf{0.015}$ |
| MLP | | | |
| Base | – | $0.745 \pm 0.034$ | $0.871 \pm 0.027$ |
| LLM-FE | Llama 3.1-8B | $0.768 \pm 0.032$ | $0.794 \pm 0.016$ |
| | GPT-3.5 Turbo | $\mathbf{0.791} \pm \mathbf{0.029}$ | $\mathbf{0.631} \pm \mathbf{0.043}$ |
| TabPFN[*] | | | |
| Base | – | $0.852 \pm 0.028$ | $0.289 \pm 0.016$ |
| LLM-FE | Llama 3.1-8B | $0.856 \pm 0.017$ | $0.288 \pm 0.016$ |
| | GPT-3.5 Turbo | $\mathbf{0.863} \pm \mathbf{0.018}$ | $\mathbf{0.286} \pm \mathbf{0.015}$ |

## 6 Conclusion

In this work, we introduce a novel framework LLM-FE that leverages LLMs as evolutionary optimizers to discover new features for tabular prediction tasks. By combining LLM-driven hypothesis generation with data-driven feedback and evolutionary search, LLM-FE effectively automates the feature engineering process. Through comprehensive experiments on diverse tabular learning tasks, we demonstrate that LLM-FE consistently outperforms state-of-the-art baselines, delivering substantial improvements in predictive performance across various tabular prediction models. Future work could explore integrating more powerful or domain-specific language models to enhance the relevance and quality of generated features for domain-specific problems. Moreover, our framework could extend beyond feature engineering to other stages of the tabular learning and data-centric pipeline, such as data augmentation, automated data cleaning (including imputation and outlier detection), and model tuning.

## Impact Statement

The introduction of LLM-FE as a framework for LLM-driven automated feature engineering can improve predictive performance while reducing manual effort, particularly in resource-intensive domains. By combining domain knowledge with evolutionary optimization, LLM-FE helps discover effective feature representations that may be difficult to engineer manually. While this work focuses on feature engineering, the framework could extend to related stages of the ML pipeline, such as data cleaning, exploratory analysis, data augmentation, and model tuning. LLM-FE may include serialized data samples in prompts; privacy considerations are important, particularly in sensitive domains. This concern is not unique to our approach, as prior LLM-based feature engineering methods also rely on prompt examples. In practice, several mitigations are possible: LLM-FE can operate with locally deployed open-source models to avoid transmitting external data, anonymized or synthetic samples can be used instead of raw data, and prompts can omit dataset examples entirely with only a minor performance impact, as shown in our ablation analysis. The automation of feature discovery is well-established in the AutoML literature, and LLM-FE continues this trajectory. LLM-FE crucially depends on user inputs such as task descriptions, feature metadata, and domain context, an example of human-in-the-loop AutoML. Rather than replacing domain experts, LLM-FE is designed to augment them, allowing data scientists to focus on higher-level problem-solving and decision-making. In safety-critical domains, LLM-FE's outputs should be treated as expert-reviewable hypotheses rather than directly deployable transformations.

## Reproducibility Statement

To ensure the reproducibility of our work, we provide comprehensive implementation details of LLM-FE. Section 3 outlines the full methodology, while Appendix B.1 offers an in-depth description of the framework, including the specific LLM prompts used. The datasets employed in our experiments are detailed in Appendix C. Additionally, we release our code and data[1] to facilitate further research.

## Acknowledgments

This research was partially supported by the U.S. National Science Foundation (NSF) under Grant No. 2416728.

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

## A    Comparison with LLM-based Baselines

While LLM-FE, CAAFE, and OCTree all leverage LLMs for automated feature engineering, they differ fundamentally in how they explore and refine the feature space. The key methodological differences are: **(i) Parallel and Multi-Path Exploration.** LLM-FE explores the feature space in parallel by evolving multiple candidate programs simultaneously across islands. CAAFE and OCTree both follow a single-path optimization process, in which the LLM incrementally refines a single candidate or rule at a time. **(ii) Population-Based Memory.** Unlike CAAFE and OCTree, LLM-FE maintains a multi-population external memory that stores diverse, high-performing feature programs across iterations. **(iii) Feedback and Refinement Design.** LLM-FE applies LLM-guided mutation and crossover over multiple populations to drive exploration and recombination. In contrast, OCTree relies on stepwise rule refinement informed by decision-tree feedback, whereas CAAFE employs prompt-based refinement without using evolutionary operators. **(iv) Empirical Impact.** The multi-island strategy helps mitigate premature convergence by preserving diversity across islands, leading to consistent empirical gains over both CAAFE and OCTree (see Tables 2 and 3). **(v) Feature Complexity**. Prior work (Küken et al., 2024) shows that LLM-based methods tend to favor simple feature operators. Our results in Figure 4 indicate that LLM-FE mitigates this bias: approximately 45% of discovered features qualify as complex under Küken et al. (2024)'s definition, compared to negligible for CAAFE and OCTree.

## B    Implementation Details

### B.1    LLM-FE

**Feature Generation.**    Figure 9 presents an example prompt for the balance-scale dataset. The prompt begins with general instructions, followed by dataset-specific details, such as task descriptions, feature descriptions, and a subset of data instances serialized and expressed in natural language. To introduce diversity in prompting, we randomly sample between this approach and an alternative set of instructions, encouraging the LLM to explore a wider range of operators from OpenFE (Zhang et al., 2023), as prior LLMs tend to favor simpler operators (Küken et al., 2024) (see Figure 8). The quality of features generated has been detailed in Appendix 5.2. By providing this structured context, the model can leverage its domain knowledge to generate semantically and contextually meaningful hypotheses for new feature optimization programs.

**Data-Driven Evaluation.**    After prompting the LLM, we sample $b = 3$ outputs. Based on preliminary experiments, we set the temperature for LLM output generation to $t = 0.8$ to balance creativity (exploration) and adherence to problem constraints, as well as reliance on prior knowledge (exploitation). The data modification process is illustrated in Figure 9(c), where the outputs are used to modify the features via `modify_features(input)`. These modified features are then input into a prediction model, and the resulting validation score is calculated. To ensure efficiency, our evaluation is constrained by time and memory limits set at $T = 30$ seconds and $M = 2GB$, respectively. Programs exceeding these limits are disqualified and assigned None scores, ensuring timely progress and resource efficiency in the search process.

**Memory Management.**    Following the 'islands' model used by Cranmer (2023); Shojaee et al. (2025); Romera-Paredes et al. (2024), we maintain the generated hypotheses along with their evaluation scores in a memory buffer comprising multiple islands ($m = 3$) that evolve independently. Each island is initialized with a simple feature transformation program specific to the dataset (`def modify_features_v0()` in Figure 9(d)). In each iteration, novel hypotheses and their validation metrics are incorporated into their respective islands only if they exceed the island's current best score. Within each island, we additionally cluster feature discovery programs based on their signature, characterized by their validation score. Feature transformation programs that produce identical scores are consolidated together, creating distinct clusters. This clustering approach helps preserve diversity by ensuring that programs with varying performance characteristics remain in the population. We leverage this island model to construct prompts for the LLM. After an initial update

of the prompt template with dataset-specific information, we integrate in-context demonstrations from the buffer. Following Shojaee et al. (2025); Romera-Paredes et al. (2024), we randomly select one of the $m$ available islands. Within the chosen island, we sample $k = 3$ programs to serve as in-context examples. To sample programs, we first select clusters based on their signatures using the Boltzmann selection strategy (De La Maza & Tidor, 1992), with a preference for higher-scoring clusters. Let $s_i$ be the score of the i-th cluster, and the probability $P_i$ for selecting the i-th cluster is:

$$P_i = \frac{exp(\frac{s_i}{\tau_c})}{\sum_i exp(\frac{s_i}{\tau_c})}, \text{ where } \tau_c = T_0(1 - \frac{u \bmod N}{N}), \tag{4}$$

where $\tau_c$ is the temperature parameter, $u$ is the current number of programs on the island, and $T_0 = 0.1$ and $N = 10,000$ are hyperparameters. Once a cluster is selected, we sample the programs from it.

```
###
<Role>
You are a data scientist with expert knowledge about the provided dataset.
Your primary responsibility is to identify the most informative features that can enhance the solution to the
specified <Task>.

###
<Instructions>
  - You are given a task description, a list of existing features, a set of advanced operators, and sample
data.
  - Your objective is to leverage the provided advanced operators within <Operators> to generate meaningful
and insightful features that enhance task performance. These operators have been carefully curated to extract
deeper patterns from the data.
  - Avoid relying on basic arithmetic operators (e.g., addition, subtraction, multiplication, or division).
Instead, focus exclusively on the provided advanced operators inside <Operators>.
  - For each feature you derive, provide a concise explanation of why it is relevant and to solving the <Task>
in the docstring.

###
<Operators>
  - General Operators: Frequency (Frequency of feature in the data)
  - Numerical Input Operators: Absolute, Logarithm, Square Root, Sigmoid, Square, Round, Residual
  - Numeric-Numeric Operators: Minimum, Maximum
  - Categorical-Numeric Operators: GroupByThenMin, GroupByThenMax, GroupByThenMean, GroupByThenMedian,
GroupByThenStd, GroupByThenRank
  - Categorical-Categorical Operators: Combine, CombineThenFreq, GroupByThenNUnique
                                                                                     Instruction
```

Figure 8: **An example of the alternate set of instructions** used to direct the model to use a complex set of operations over simple operators for generating features.

## B.2   Baselines

We implement and evaluate various state-of-the-art feature engineering baselines, spanning traditional methods to recent LLM-based approaches, for comparison with LLM-FE. After generating features with each baseline, we apply a unified preprocessing pipeline to prepare the data for training and evaluation in the machine learning model. We implement the following baselines:

**FeatLLM.**   FeatLLM uses an LLM to generate rules to binarize features that are then used as input to a simple model, such as linear regression. We adapt the open-source `featllm`[2] implementation, modifying the pipeline to use an `XGBoost` model for inference. To ensure a fair comparison with other methods, we train the `XGBoost` model on the entire training dataset, while the LLM uses only a subset (10 samples) to generate binary features. As `FeatLLM` generates multiple feature sets in parallel across LLM calls, we report the results through an ensemble over three samples to maintain consistency with LLM-FE.

**AutoFeat.**   AutoFeat is a classical feature engineering approach that uses iterative feature subsampling with beam search to select informative features. We utilize the open-source `autofeat`[3] package, retaining the default parameter settings. For parameter settings, we refer to the example '.ipynb' files provided in their official repository.

---

[2]https://github.com/Sungwon-Han/FeatLLM
[3]https://github.com/cod3licious/autofeat.git

```
###
<Role>
You are a data scientist expert in the field of the given dataset.
Your role is to apply your domain expertise to identify and create, and refine the most informative features
that solve the <Task> effectively.

###
<Instructions>
- You are provided with the task description, a list of existing features, and data examples.
- Use your domain knowledge to derive features that capture meaningful patterns, trends, or relationships
inherent in the data.
- Prioritize features that have high potential to enhance the model's ability to solve the <Task>, considering
both relevance and predictive power.
- For each derived feature, provide:
- A clear explanation of how it was derived and justification of its relevance for solving the <Task>.
- Ensure your approach remains grounded in the context of the dataset and the <Task>, and aim for features
that are interpretable and actionable.
```
**Instruction**

```
###
<Task>
Which direction does the balance scale tip to? Right, left, or balanced?

###
<Features>
- Left-Weight: Left-Weight (numerical variable within range [1, 5])
....
....

###
<Examples>
If Left-Weight is 3, Left-Distance is 3, Right-Weight is 4, Right-Distance is 5,  Then Result is right.
....
....

Please generate as many new features as possible using the information from the task, feature descriptions,
examples, and your domain understanding of the dataset. Remove any irrelevant, redundant, or less informative
features to enhance overall performance.

First, describe your new feature transformation and the main steps in a concise, one-sentence docstring.Then,
implement it in Python as a function that adheres to the given specifications.
Avoid providing any further explanations or additional descriptions.
```
**Dataset Specification**

```python
def evaluate(data: dict):
    """ Evaluate the feature transformations on data observations."""
    import torch
    import utils
    from sklearn.model_selection import train_test_split
    from sklearn.metrics import accuracy_score
    from sklearn import preprocessing
    import xgboost as xgb

    #Data Loading and Processing

    # Load model
    model = xgb.XGBClassifier(random_state=42)
    # Training
    model.fit(X_train, y_train)
    # Inference
    y_pred = model.predict(X_test)
    score = accuracy_score(y_test, y_pred)

    return score, inputs, outputs
```
**Evaluation Function**

```python
# Load data observations
label_encoder = preprocessing.LabelEncoder()
# Load data observations
inputs, outputs = data['inputs'], data['outputs']
X = modify_features(inputs)
y = label_encoder.fit_transform(outputs)
for col in X.columns:
    if X[col].dtype == 'string':
        X[col] = label_encoder.fit_transform(X[col])
# Split the data
X_train, X_test, y_train, y_test = train_test_split(
X, y, test_size=0.25, random_state=0)
# Data Processing
X_train = utils.make_numeric(X_train)
X_test = utils.make_numeric(X_test)

X_train = torch.tensor(X_train.to_numpy())
X_test = torch.tensor(X_test.to_numpy())
```

```python
def modify_features_v0(df_input) -> pd.DataFrame:
    """
    Thought 1: The absolute difference between Left-Weight and Right-Weight can
    capture the imbalance in weight distribution.
    Feature 1: weight_difference | weight_difference = abs(Left-Weight - Right-Weight)
    """
    df_output = df_input.copy()
    # Calculate absolute difference between Left-Weight and Right-Weight
    df_output['weight_difference'] =
                abs(df_output['Left-Weight'] - df_output['Right-Weight'])

    return df_output
```
**In-Context Example**

```python
def modify_features_v1(df_input) -> pd.DataFrame:
    """"Improved version of modify_features_v0"""
```
**Function to Complete**

Figure 9: **Example of an input prompt for balance-scale dataset** containing (a) instruction, (b) dataset specification containing the details about the task, features, and data samples, (c) evaluation function, (d) initial in-context demonstration, and (e) function to complete.

**CAAFE.** We utilize the official implementation of `CAAFE`,[4], maintaining all parameter settings as specified in the original repository. Additionally, the repository is designed for classification datasets. Following their workflow, we preprocess the data before feeding it into the prediction model after feature engineering. As `CAAFE` implements sequential feature refinement and produces only a single independent candidate solution, ensembling is not applicable.

**OCTree.** The official `OCTree` implementation[5] was modified to keep the data loading and model initialization part common. We implemented OCTree only for classification datasets, as the official implementation is limited to classification datasets, and running for regression datasets on our own could have resulted in incorrect implementation. Furthermore, `OCTree` also follows a sequential optimization procedure and does not produce multiple independent solutions for ensembling.

**OpenFE.** OpenFE is another state-of-the-art traditional feature engineering method using feature boosting and pruning algorithms. We employ the open-source `openfe`[6] package with standard parameter settings.

## C  Dataset Details

Table 6 describes the diverse collection of datasets spanning three major categories: (1) binary classification, (2) multi-class classification, and (3) regression problems used in our evaluation. The datasets were primarily sourced from established platforms, including OpenML (Vanschoren et al., 2014; Feurer et al., 2021), UCI (Asuncion et al., 2007), and Kaggle. We specifically selected datasets with descriptive feature names, excluding those with merely numerical identifiers. Each dataset includes a task description, enhancing users' contextual understanding. Our selection encompasses not only small datasets but also larger ones, including extensive datasets and high-dimensional ones with over 50 features. This diverse and comprehensive selection of datasets spans a broad spectrum of real-world scenarios, with varying feature dimensionality and sample sizes, thereby providing a robust framework for evaluating feature engineering methods.

Table 6: Dataset statistics.

| Dataset | #Features | #Samples | Source | ID/Name |
|---|---|---|---|---|
| Binary Classification | | | | |
| adult | 14 | 48842 | OpenML | 1590 |
| blood-transfusion | 4 | 748 | OpenML | 1464 |
| bank-marketing | 16 | 45211 | OpenML | 1461 |
| breast-w | 9 | 699 | OpenML | 15 |
| credit-g | 20 | 1000 | OpenML | 31 |
| tic-tac-toe | 9 | 958 | OpenML | 50 |
| pc1 | 21 | 1109 | OpenML | 1068 |
| Multi-class Classification | | | | |
| arrhythmia | 279 | 452 | OpenML | 5 |
| balance-scale | 4 | 625 | OpenML | 11 |
| car | 6 | 1728 | OpenML | 40975 |
| cmc | 9 | 1473 | OpenML | 23 |
| eucalyptus | 19 | 736 | OpenML | 188 |
| jungle_chess | 6 | 44819 | OpenML | 41027 |
| vehicle | 18 | 846 | OpenML | 54 |
| cdc diabetes | 21 | 253680 | Kaggle | diabetes-health-indicators-dataset |
| heart | 11 | 918 | Kaggle | heart-failure-prediction |
| communities | 103 | 1994 | UCI | communities-and-crime |
| covtype | 54 | 581012 | UCI | covertype |
| myocardial | 111 | 1700 | UCI | myocardial-infarction-complications |
| Regression | | | | |
| airfoil_self_noise | 6 | 1503 | OpenML | 44957 |
| cpu_small | 12 | 8192 | OpenML | 562 |
| diamonds | 9 | 53940 | OpenML | 42225 |
| plasma_retinol | 13 | 315 | OpenML | 511 |
| forest-fires | 13 | 517 | OpenML | 42363 |
| housing | 9 | 20640 | OpenML | 43996 |
| crab | 8 | 3893 | Kaggle | crab-age-prediction |
| insurance | 7 | 1338 | Kaggle | us-health-insurancedataset |
| bike | 11 | 17389 | UCI | bike-sharing-dataset |
| wine | 10 | 4898 | UCI | wine-quality |

---

[4] https://github.com/noahho/CAAFE

[5] https://github.com/jaehyun513/OCTree

[6] https://github.com/IIIS-Li-Group/OpenFE.git

## D  Additional Results

### D.1  LLM-FE and Hyperparameter Optimization (HPO)

To assess the impact of hyperparameter optimization (HPO) on LLM-FE, we conduct experiments with XGBoost and Multilayer Perceptron (MLP) models across five classification datasets where baseline models achieve accuracies below 0.8. We adopt the hyperparameter search spaces detailed in Table 7 (XGBoost) and Table 8 (MLP), following prior work (Grinsztajn et al., 2022; Gorishniy et al., 2021). Optimization is performed with `Optuna` (Akiba et al., 2019), using 100 trials with random sampling across multiple dataset splits. All MLP models are trained for up to 100 epochs with early stopping, retaining the checkpoint that achieves the best validation score.

Table 7: XGBoost hyperparameters space.

| Parameter | Distribution |
|---|---|
| Max depth | UniformInt [1, 11] |
| Num estimators | UniformInt [100, 6100, 200] |
| Min child weight | LogUniformInt [1, 1e2] |
| Subsample | Uniform [0.5, 1] |
| Learning rate | LogUniform [1e-5, 0.7] |
| Col sample by level | Uniform [0.5, 1] |
| Col sample by tree | Uniform [0.5, 1] |
| Gamma | LogUniform [1e-8, 7] |
| Lambda | LogUniform [1, 4] |
| Alpha | LogUniform [1e-8, 1e2] |

Table 8: MLP hyperparameters space.

| Parameter | Distribution |
|---|---|
| Num layers | UniformInt [1, 8] |
| Layer size | UniformInt [16, 1024] |
| Dropout | Uniform [0, 0.5] |
| Learning rate | LogUniform [1e-5, 1e-2] |
| Category embedding size | UniformInt [64, 512] |
| Learning rate scheduler | {True, False} |
| Batch size | {256, 512, 1024} |

As summarized in Table 9, we compare the performance of LLM-FE with base and OpenFE using HPO. HPO consistently improves performance across all datasets for the Base model. Crucially, our proposed method LLM-FE delivers further gains over the base model and OpenFE in three of the five datasets even after HPO. These results show that while HPO provides meaningful improvements, LLM-FE offers complementary, substantial enhancements that are independent of hyperparameter tuning.

Table 9: **Comparison of classification accuracy across datasets using XGBoost and MLP models. bold**: indicates best performance.

| Dataset | XGBoost | | | MLP | | |
|---|---|---|---|---|---|---|
| | Base | OpenFE | LLM-FE | Base | OpenFE | LLM-FE |
| eucalyptus | $0.681 \pm 0.029$ | $\mathbf{0.687 \pm 0.017}$ | $0.678 \pm 0.020$ | $0.501 \pm 0.041$ | $0.376 \pm 0.080$ | $\mathbf{0.506 \pm 0.028}$ |
| credit-g | $0.746 \pm 0.023$ | $0.754 \pm 0.019$ | $\mathbf{0.755 \pm 0.020}$ | $0.689 \pm 0.032$ | $0.643 \pm 0.047$ | $\mathbf{0.693 \pm 0.028}$ |
| cmc | $0.552 \pm 0.030$ | $0.551 \pm 0.013$ | $\mathbf{0.560 \pm 0.030}$ | $\mathbf{0.572 \pm 0.024}$ | $0.491 \pm 0.023$ | $0.567 \pm 0.027$ |
| blood-transfusion | $0.790 \pm 0.010$ | $0.777 \pm 0.016$ | $\mathbf{0.791 \pm 0.011}$ | $0.616 \pm 0.182$ | $\mathbf{0.746 \pm 0.031}$ | $0.705 \pm 0.078$ |
| vehicle | $0.760 \pm 0.016$ | $\mathbf{0.810 \pm 0.016}$ | $0.780 \pm 0.022$ | $0.637 \pm 0.095$ | $0.396 \pm 0.043$ | $\mathbf{0.694 \pm 0.039}$ |

### D.2  Generalizability Analysis

We extend the results from Section 4.4 to showcase the performance improvements achieved by LLM-FE across various prediction models. Specifically, we employ `XGBoost`, `MLP`, and `TabPFN` to generate features and subsequently use the same models for inference. As shown in Table 10, the features generated with `GPT-3.5-Turbo` by LLM-FE consistently improve model performance across datasets, outperforming base versions trained without feature engineering. To further assess the generalizability of LLM-FE, we conducted experiments on three additional LLMs (`GPT-4o-mini` (OpenAI, 2023), `Qwen2.5-72B-Instruct` (Yang et al., 2024a), and `Gemini-2.5-Flash` (Comanici et al., 2025)) (Table 11) and smaller prediction models like CatBoost and Logistic Regression (Table 12). From Tables 10, 11, and 12, LLM-FE outperforms the respective base models for most of the datasets.

Table 10: **Performance improvement with LLM-FE.** We report the mean and standard deviation over five splits. We use RMSE for regression datasets (lower values indicate better performance) and Accuracy for classification datasets (higher values indicate better performance). **bold:** indicates the best performance.

| Dataset | XGBoost | | MLP | | TabPFN | |
|---|---|---|---|---|---|---|
| | Base | LLM-FE | Base | LLM-FE | Base | LLM-FE |
| Classification Datasets | | | | | | |
| breast-w | $0.956 \pm 0.012$ | $\mathbf{0.973} \pm \mathbf{0.009}$ | $0.957 \pm 0.010$ | $\mathbf{0.964} \pm \mathbf{0.005}$ | $\mathbf{0.971} \pm \mathbf{0.006}$ | $\mathbf{0.971} \pm \mathbf{0.007}$ |
| blood-transfusion | $0.742 \pm 0.012$ | $\mathbf{0.751} \pm \mathbf{0.036}$ | $0.674 \pm 0.071$ | $\mathbf{0.782} \pm \mathbf{0.017}$ | $0.790 \pm 0.012$ | $\mathbf{0.791} \pm \mathbf{0.011}$ |
| car | $0.995 \pm 0.003$ | $\mathbf{0.999} \pm \mathbf{0.001}$ | $0.929 \pm 0.019$ | $\mathbf{0.950} \pm \mathbf{0.009}$ | $0.984 \pm 0.007$ | $\mathbf{0.996} \pm \mathbf{0.006}$ |
| cmc | $0.528 \pm 0.030$ | $\mathbf{0.535} \pm \mathbf{0.019}$ | $0.559 \pm 0.028$ | $\mathbf{0.566} \pm \mathbf{0.028}$ | $0.563 \pm 0.030$ | $\mathbf{0.566} \pm \mathbf{0.036}$ |
| credit-g | $0.751 \pm 0.019$ | $\mathbf{0.766} \pm \mathbf{0.025}$ | $0.558 \pm 0.144$ | $\mathbf{0.633} \pm \mathbf{0.101}$ | $0.728 \pm 0.008$ | $\mathbf{0.794} \pm \mathbf{0.022}$ |
| eucalyptus | $0.655 \pm 0.024$ | $\mathbf{0.668} \pm \mathbf{0.027}$ | $0.414 \pm 0.064$ | $\mathbf{0.456} \pm \mathbf{0.062}$ | $0.712 \pm 0.016$ | $\mathbf{0.715} \pm \mathbf{0.021}$ |
| heart | $0.858 \pm 0.013$ | $\mathbf{0.866} \pm \mathbf{0.021}$ | $0.840 \pm 0.010$ | $\mathbf{0.844} \pm \mathbf{0.006}$ | $\mathbf{0.882} \pm \mathbf{0.025}$ | $0.880 \pm 0.021$ |
| pc1 | $0.931 \pm 0.004$ | $\mathbf{0.935} \pm \mathbf{0.006}$ | $\mathbf{0.931} \pm \mathbf{0.002}$ | $0.904 \pm 0.055$ | $0.936 \pm 0.007$ | $\mathbf{0.937} \pm \mathbf{0.003}$ |
| vehicle | $0.754 \pm 0.016$ | $\mathbf{0.769} \pm \mathbf{0.027}$ | $0.583 \pm 0.062$ | $\mathbf{0.673} \pm \mathbf{0.043}$ | $0.852 \pm 0.016$ | $\mathbf{0.856} \pm \mathbf{0.028}$ |
| Regression Datasets | | | | | | |
| bike $[10^1]$ | $4.094 \pm 0.096$ | $\mathbf{3.985} \pm \mathbf{0.084}$ | $1.205 \pm 0.028$ | $\mathbf{1.044} \pm \mathbf{0.042}$ | $3.795 \pm 0.094$ | $\mathbf{3.759} \pm \mathbf{0.109}$ |
| crab $[10^0]$ | $2.325 \pm 0.094$ | $\mathbf{2.211} \pm \mathbf{0.124}$ | $2.128 \pm 0.104$ | $\mathbf{2.110} \pm \mathbf{0.106}$ | $2.073 \pm 0.115$ | $\mathbf{2.065} \pm \mathbf{0.134}$ |
| housing $[10^4]$ | $4.845 \pm 0.191$ | $\mathbf{4.525} \pm \mathbf{0.260}$ | $1.045 \pm 0.018$ | $9.183 \pm 0.734$ | $4.338 \pm 0.081$ | $\mathbf{4.184} \pm \mathbf{0.071}$ |
| insurance $[10^3]$ | $5.269 \pm 0.260$ | $\mathbf{5.069} \pm \mathbf{0.392}$ | $1.189 \pm 0.070$ | $6.459 \pm 0.341$ | $4.653 \pm 0.237$ | $\mathbf{4.592} \pm \mathbf{0.261}$ |
| wine $[10^0]$ | $0.639 \pm 0.006$ | $\mathbf{0.612} \pm \mathbf{0.007}$ | $0.728 \pm 0.005$ | $\mathbf{0.728} \pm \mathbf{0.003}$ | $0.678 \pm 0.023$ | $\mathbf{0.676} \pm \mathbf{0.024}$ |

Table 11: **Generalization across LLMs.** Performance of LLM-FE using different LLM backbones using the XGBoost model. We use RMSE for regression datasets (lower values indicate better performance) and Accuracy for classification datasets (higher values indicate better performance). **bold:** indicates the best performance.

| Dataset | Base | Qwen2.5-72B | GPT-4o-mini | Gemini-2.5-Flash |
|---|---|---|---|---|
| Classification Datasets | | | | |
| breast-w | $0.956 \pm 0.012$ | $\mathbf{0.974} \pm \mathbf{0.006}$ | $0.969 \pm 0.011$ | $0.961 \pm 0.011$ |
| blood-transfusion | $0.742 \pm 0.012$ | $\mathbf{0.750} \pm \mathbf{0.029}$ | $0.749 \pm 0.022$ | $0.747 \pm 0.023$ |
| car | $0.995 \pm 0.003$ | $\mathbf{1.000} \pm \mathbf{0.000}$ | $0.999 \pm 0.001$ | $0.997 \pm 0.005$ |
| cmc | $0.528 \pm 0.029$ | $0.534 \pm 0.016$ | $\mathbf{0.538} \pm \mathbf{0.016}$ | $0.534 \pm 0.024$ |
| credit-g | $0.751 \pm 0.019$ | $\mathbf{0.775} \pm \mathbf{0.022}$ | $0.764 \pm 0.028$ | $0.755 \pm 0.012$ |
| eucalyptus | $0.655 \pm 0.024$ | $\mathbf{0.678} \pm \mathbf{0.028}$ | $0.670 \pm 0.022$ | $0.659 \pm 0.032$ |
| heart | $0.858 \pm 0.013$ | $\mathbf{0.863} \pm \mathbf{0.023}$ | $0.857 \pm 0.016$ | $0.847 \pm 0.012$ |
| pc1 | $0.931 \pm 0.004$ | $0.934 \pm 0.004$ | $\mathbf{0.935} \pm \mathbf{0.008}$ | $0.934 \pm 0.009$ |
| vehicle | $0.754 \pm 0.016$ | $\mathbf{0.770} \pm \mathbf{0.020}$ | $0.761 \pm 0.027$ | $0.766 \pm 0.026$ |
| Regression Datasets | | | | |
| bike $[10^1]$ | $4.094 \pm 0.096$ | $4.027 \pm 0.322$ | $\mathbf{3.928} \pm \mathbf{0.186}$ | $3.960 \pm 0.119$ |
| crab $[10^0]$ | $2.325 \pm 0.094$ | $2.262 \pm 0.203$ | $\mathbf{2.193} \pm \mathbf{0.132}$ | $2.273 \pm 0.154$ |
| housing $[10^4]$ | $4.845 \pm 0.191$ | $4.813 \pm 0.344$ | $\mathbf{4.430} \pm \mathbf{0.126}$ | $4.769 \pm 0.393$ |
| insurance $[10^3]$ | $5.269 \pm 0.260$ | $5.108 \pm 0.296$ | $\mathbf{5.100} \pm \mathbf{0.351}$ | $5.351 \pm 0.776$ |
| wine $[10^0]$ | $0.639 \pm 0.006$ | $\mathbf{0.610} \pm \mathbf{0.004}$ | $0.616 \pm 0.006$ | $0.633 \pm 0.005$ |

Table 12: **Performance improvement with LLM-FE on CatBoost and Logistic Regression.** We report the mean and standard deviation over five splits. We use Accuracy for classification datasets, with a higher value indicating better performance. **bold:** indicates the best performance.

| Dataset | Logistic Regression | | CatBoost | |
|---|---|---|---|---|
| | Base | LLM-FE | Base | LLM-FE |
| breast-w | $0.955 \pm 0.014$ | $\mathbf{0.962} \pm \mathbf{0.008}$ | $0.957 \pm 0.009$ | $\mathbf{0.962} \pm \mathbf{0.008}$ |
| blood-transfusion | $0.799 \pm 0.014$ | $\mathbf{0.799} \pm \mathbf{0.009}$ | $0.742 \pm 0.012$ | $\mathbf{0.751} \pm \mathbf{0.036}$ |
| car | $0.690 \pm 0.017$ | $\mathbf{0.696} \pm \mathbf{0.031}$ | $0.999 \pm 0.001$ | $\mathbf{0.999} \pm \mathbf{0.001}$ |
| cmc | $0.520 \pm 0.019$ | $\mathbf{0.525} \pm \mathbf{0.012}$ | $0.518 \pm 0.028$ | $\mathbf{0.548} \pm \mathbf{0.027}$ |
| credit-g | $0.764 \pm 0.006$ | $\mathbf{0.780} \pm \mathbf{0.015}$ | $\mathbf{0.714} \pm \mathbf{0.046}$ | $0.700 \pm 0.021$ |
| eucalyptus | $\mathbf{0.671} \pm \mathbf{0.036}$ | $0.667 \pm 0.042$ | $0.436 \pm 0.027$ | $\mathbf{0.509} \pm \mathbf{0.050}$ |
| heart | $\mathbf{0.877} \pm \mathbf{0.021}$ | $0.872 \pm 0.025$ | $\mathbf{0.845} \pm \mathbf{0.015}$ | $0.839 \pm 0.018$ |
| pc1 | $0.931 \pm 0.003$ | $\mathbf{0.935} \pm \mathbf{0.003}$ | $0.929 \pm 0.005$ | $\mathbf{0.932} \pm \mathbf{0.012}$ |
| vehicle | $\mathbf{0.772} \pm \mathbf{0.028}$ | $0.769 \pm 0.015$ | $0.719 \pm 0.045$ | $\mathbf{0.725} \pm \mathbf{0.033}$ |

### D.3 Transferability of Generated Features

While traditional approaches typically use the same model for both feature generation and inference, we demonstrate that the features generated by one model can be utilized by other models. Following Nam et al. (2024), we use `XGBoost`, a computationally efficient decision tree-based model, to generate features for more complex architectures during inference. As demonstrated in Table 13, `XGBoost`-generated features show an improvement in the performance of `MLP` and `TabPFN` over their base versions. This cross-architecture performance improvement suggests that the generated features capture meaningful data characteristics that are valuable across different modeling paradigms.

Table 13: **Comparative analysis of LLM-FE using feature transfer**. We report classification accuracy and normalized root-mean-square error for regression tasks. Mean and standard deviation across five random splits. **bold:** indicates best performance.

| Method | LLM | Classification ↑ | Regression ↓ |
|---|---|---|---|
| | | MLP | |
| Base | – | $0.745 \pm 0.034$ | $0.871 \pm 0.027$ |
| **LLM-FE**$_{\text{XGB}}$ | GPT-3.5-Turbo | $0.763 \pm 0.030$ | $0.848 \pm 0.017$ |
| **LLM-FE** | GPT-3.5-Turbo | $\mathbf{0.791 \pm 0.029}$ | $\mathbf{0.631 \pm 0.043}$ |
| | | TabPFN | |
| Base | – | $0.852 \pm 0.028$ | $0.289 \pm 0.016$ |
| **LLM-FE**$_{\text{XGB}}$ | GPT-3.5-Turbo | $0.861 \pm 0.017$ | $0.287 \pm 0.015$ |
| **LLM-FE** | GPT-3.5-Turbo | $\mathbf{0.863 \pm 0.018}$ | $\mathbf{0.286 \pm 0.015}$ |

### D.4 Statistical Analysis

To evaluate whether feature engineering provides statistically significant improvements over the raw feature set, we conduct one-tailed Wilcoxon signed-rank tests comparing LLM-FE against the base XGBoost model across all datasets for both classification and regression tasks (Table 14). The one-tailed formulation is appropriate as our hypothesis is strictly directional: feature engineering should improve predictive performance over the raw feature set. Tests were conducted across 25 paired observations per dataset, obtained from 5 random seeds across 5-fold cross-validation. LLM-FE achieves statistically significant improvement (using 95% confidence intervals) over the base model on all 10 out of 10 regression datasets ($p < 0.001$) and 8 out of 15 classification datasets ($p < 0.05$), along with marginal improvements over 2 classification datasets. Although not every dataset shows a statistically significant improvement, the consistent gains across most regression and several classification datasets suggest that LLM-guided feature engineering can yield meaningful improvements over the raw feature space.

Table 14: Wilcoxon one-tailed significance test. **bold:** statistically significant improvements. underline: marginally significant improvements.

| Dataset | Base | LLM-FE |
|---|---|---|
| | Classification | |
| adult | $0.872 \pm 0.002$ | $\mathbf{0.874 \pm 0.003}$ |
| bank | $0.906 \pm 0.002$ | $\mathbf{0.907 \pm 0.003}$ |
| breast-w | $0.955 \pm 0.014$ | $\mathbf{0.963 \pm 0.012}$ |
| blood | $0.747 \pm 0.023$ | $0.743 \pm 0.024$ |
| car | $0.992 \pm 0.005$ | $\mathbf{0.998 \pm 0.004}$ |
| cdc-diabetes | $0.849 \pm 0.001$ | $\mathbf{0.849 \pm 0.001}$ |
| cmc | $0.527 \pm 0.029$ | $0.527 \pm 0.024$ |
| communities | $0.699 \pm 0.020$ | $0.703 \pm 0.019$ |
| covtype | $0.871 \pm 0.002$ | $\underline{0.878 \pm 0.001}$ |
| credit-g | $0.754 \pm 0.028$ | $0.758 \pm 0.027$ |
| eucalyptus | $0.659 \pm 0.029$ | $\mathbf{0.671 \pm 0.029}$ |
| heart | $0.861 \pm 0.022$ | $0.857 \pm 0.026$ |
| myocardial | $0.785 \pm 0.025$ | $0.788 \pm 0.029$ |
| pc1 | $0.931 \pm 0.010$ | $\mathbf{0.937 \pm 0.008}$ |
| vehicle | $0.761 \pm 0.028$ | $0.767 \pm 0.030$ |
| | Regression | |
| forest-fires $[10^0]$ | $1.649 \pm 0.116$ | $\mathbf{1.567 \pm 0.116}$ |
| housing $[10^4]$ | $4.801 \pm 0.118$ | $\mathbf{4.422 \pm 0.144}$ |
| insurance $[10^3]$ | $5.280 \pm 0.306$ | $\mathbf{5.117 \pm 0.353}$ |
| bike $[10^1]$ | $4.078 \pm 0.122$ | $\mathbf{3.976 \pm 0.137}$ |
| wine $[10^{-1}]$ | $6.370 \pm 0.190$ | $\mathbf{6.130 \pm 0.190}$ |
| crab $[10^0]$ | $2.309 \pm 0.092$ | $\mathbf{2.215 \pm 0.097}$ |
| diamond $[10^2]$ | $5.482 \pm 0.104$ | $\mathbf{5.365 \pm 0.131}$ |
| airfoil_self_noise $[10^0]$ | $1.547 \pm 0.127$ | $\mathbf{1.435 \pm 0.113}$ |
| cpu_small $[10^0]$ | $2.833 \pm 0.210$ | $\mathbf{2.718 \pm 0.219}$ |
| plasma_retinol $[10^2]$ | $2.336 \pm 0.238$ | $\mathbf{2.240 \pm 0.268}$ |

## E Qualitative Analysis

### E.1 Interpretability Analysis

As illustrated in Figure 11, LLM-FE generates feature-transformation programs in natural language, thus supporting interpretability. Each generated feature program is evaluated independently, and successful ones are stored for evolutionary refinement, enabling early discoveries to compose into higher-order features while preserving interpretability. To evaluate the utility of the generated features, we conduct attribution analysis using SHAP values. The results demonstrate that a consistent subset of discovered features receives high attribution scores, indicating

Table 15: Percentage of generated features ranked among the top-$k$ most impactful features by SHAP.

| Top-$k$ | Percentage |
|---|---|
| Top-10 | 16.67 |
| Top-20 | 25.93 |
| Top-30 | 37.04 |
| Top-40 | 57.41 |
| Top-50 | 62.96 |

that they actively contribute to the prediction process rather than serving as spurious or unused augmentations. Specifically, 16.67% of generated features rank among the top-10 most impactful features, and over 60% appear within the top-50 (Table 15), providing strong evidence that the features discovered by LLM-FE meaningfully enhance model performance and decision-making.

## E.2 Robustness to Noise

Noise is a pervasive challenge in real-world datasets, stemming from sensor imperfections, human errors, environmental variability, and hardware constraints. Such corruption can obscure meaningful structure and hinder a model's ability to learn true underlying relationships. To assess how well LLM-FE leverages prior knowledge and evolutionary search to remain effective under noisy conditions, we added Gaussian noise ($\sigma = 0, 0.01, 0.05, 0.1$) to numerical classification datasets. As shown in Figure 10, we evaluated `XGBoost` with several feature engineering approaches, using `GPT-3.5-Turbo` as the backbone for all LLM-based methods. Across all noise levels, LLM-FE consistently maintains higher accuracy and exhibits greater robustness than competing approaches, demonstrating its resilience to noise-induced degradation.

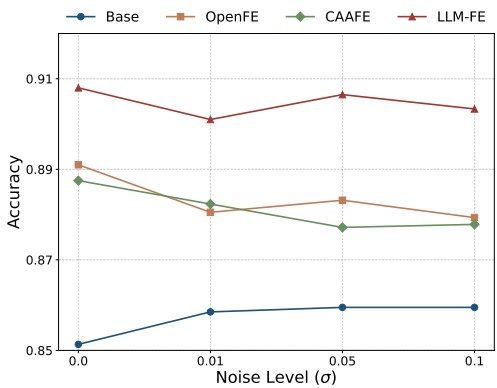

Figure 10: **Impact of Noise Levels** on XGBoost model performance across different approaches under increasing noise conditions ($\sigma \in \{0.0, 0.1\}$). We report the mean accuracy across six classification datasets containing only numerical features.

## E.3 Impact of Domain Knowledge

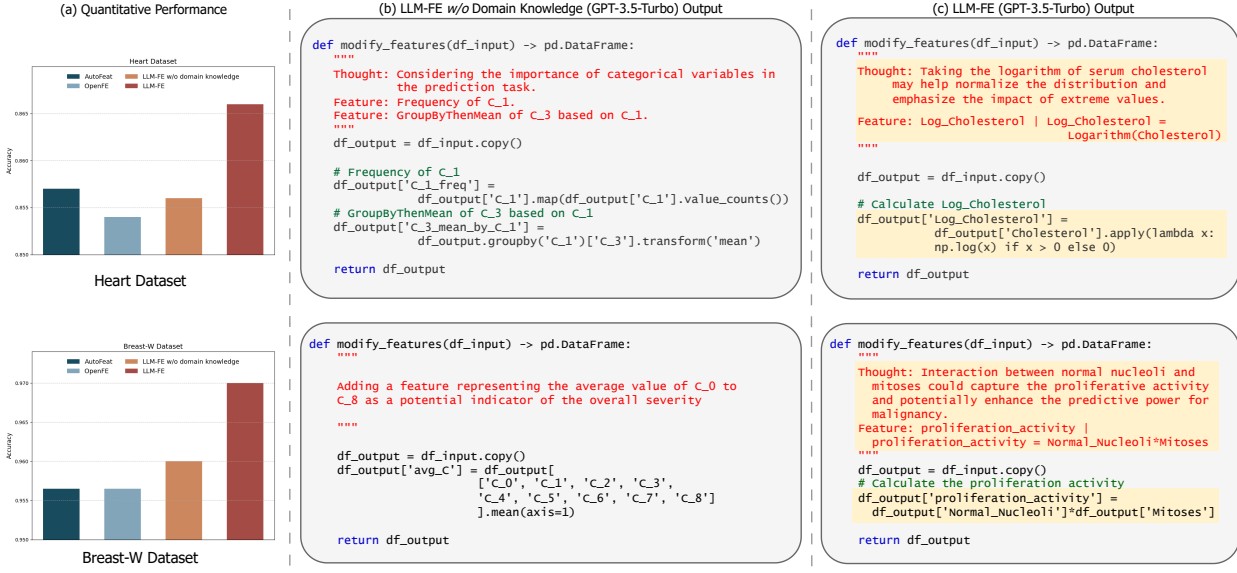

Figure 11: **Quantitative and Qualitative Analysis on Impact of Domain Knowledge for LLM-FE on Heart and Breast-W datasets.** (a) Comparison of XGBoost performance for LLM-FE against its domain-agnostic variant and traditional methods, such as OpenFE and AutoFeat, which do not integrate domain knowledge and exhibit reduced performance. (b) Features generated using the *w/o* Domain Knowledge variant of LLM-FE. (c) Feature discovery program generated by LLM-FE. The generated programs emphasize how incorporating domain expertise leads to more interpretable features that improve model performance.

Figure 11 highlights the qualitative and quantitative benefits of domain-specific feature transforms. We demonstrate this using two datasets: the Breast-W dataset, which focuses on distinguishing between benign and malignant tumors, and the Heart dataset, which predicts cardiovascular disease risk based on patient attributes. These tasks underscore the crucial role of domain knowledge in identifying meaningful features. By leveraging embedded domain knowledge, LLM-FE not only significantly improves accuracy but also provides justification for selecting the chosen feature, leading to more interpretable feature engineering. For example, in the Heart dataset, LLM-FE suggests the feature '`Log_Cholesterol`', recognizing cholesterol's critical role in heart health and applying a logarithmic transformation to reduce the impact of outliers and stabilize the variance. In contrast, the '*w/o* Domain Knowledge' variant arbitrarily combines existing features, leading to uninterpretable transformations and reduced overall performance (Figure 11(a)). Similarly, for breast cancer prediction, LLM-FE identifies '`proliferation_activity`', a biologically relevant metric leading to performance improvement, whereas the absence of domain knowledge results in a simple mean of all features, lacking interpretability and clinical significance (Figures 11(b) and 11(c)).

### E.4 Impact of Multi-Island Evolution

At initialization, the feature discovery process is partitioned into $k$ independent islands by evenly splitting the candidate feature set. Given a fixed computational budget of $T$ iterations, each island receives approximately $T/k$ iterations, making $k$ a key factor in controlling the trade-off between exploration and exploitation. Smaller values of $k$ allow deeper refinement within each island, emphasizing exploitation, while larger values of $k$ encourage broader exploration by enabling multiple independent search trajectories with shallower refinement. As shown in Table 16, we evaluate representative settings with $k = 1, 3$ and 5. Moderate island counts consistently provide the best balance between exploration and refinement. Using a single island limits the diversity of discovered features, whereas too many islands reduces the refinement depth available to each trajectory. Overall performance is robust across island counts, with independent trajectories enabling complementary exploration and stable results when the compute budget is appropriately distributed.

Table 16: **Effect of the number of islands in LLM-FE**. We report the mean and standard deviation over five splits using accuracy for classification datasets (higher is better). **bold**: indicates the best performance.

| # Islands | adult | bank | cmc | car | breast-w | vehicle |
|---|---|---|---|---|---|---|
| 1 | **0.874 ± 0.002** | **0.908 ± 0.003** | 0.532 ± 0.017 | 0.999 ± 0.003 | 0.966 ± 0.014 | 0.761 ± 0.012 |
| 3 | **0.874 ± 0.002** | 0.907 ± 0.002 | **0.535 ± 0.019** | **0.999 ± 0.001** | **0.973 ± 0.009** | 0.769 ± 0.013 |
| 5 | 0.874 ± 0.003 | 0.907 ± 0.003 | 0.528 ± 0.010 | 0.998 ± 0.003 | 0.969 ± 0.014 | **0.773 ± 0.015** |

### E.5 Impact of Evolutionary Refinement

Figure 12 shows the detailed performance trajectory of LLM-FE compared with its '*w/o* Evolutionary Refinement' variant on PC1 and Balance-Scale datasets. The graph demonstrates that LLM-FE, using evolutionary search, consistently improves validation accuracy, while the non-refinement variant stagnates due to local optima. On the PC1 dataset, the non-refinement variant plateaus after seven iterations, and on the Balance-Scale dataset, it stagnates after five iterations. LLM-FE's evolutionary refinement helps it escape local optima with more robust optimization, leading to better validation accuracy on both datasets.

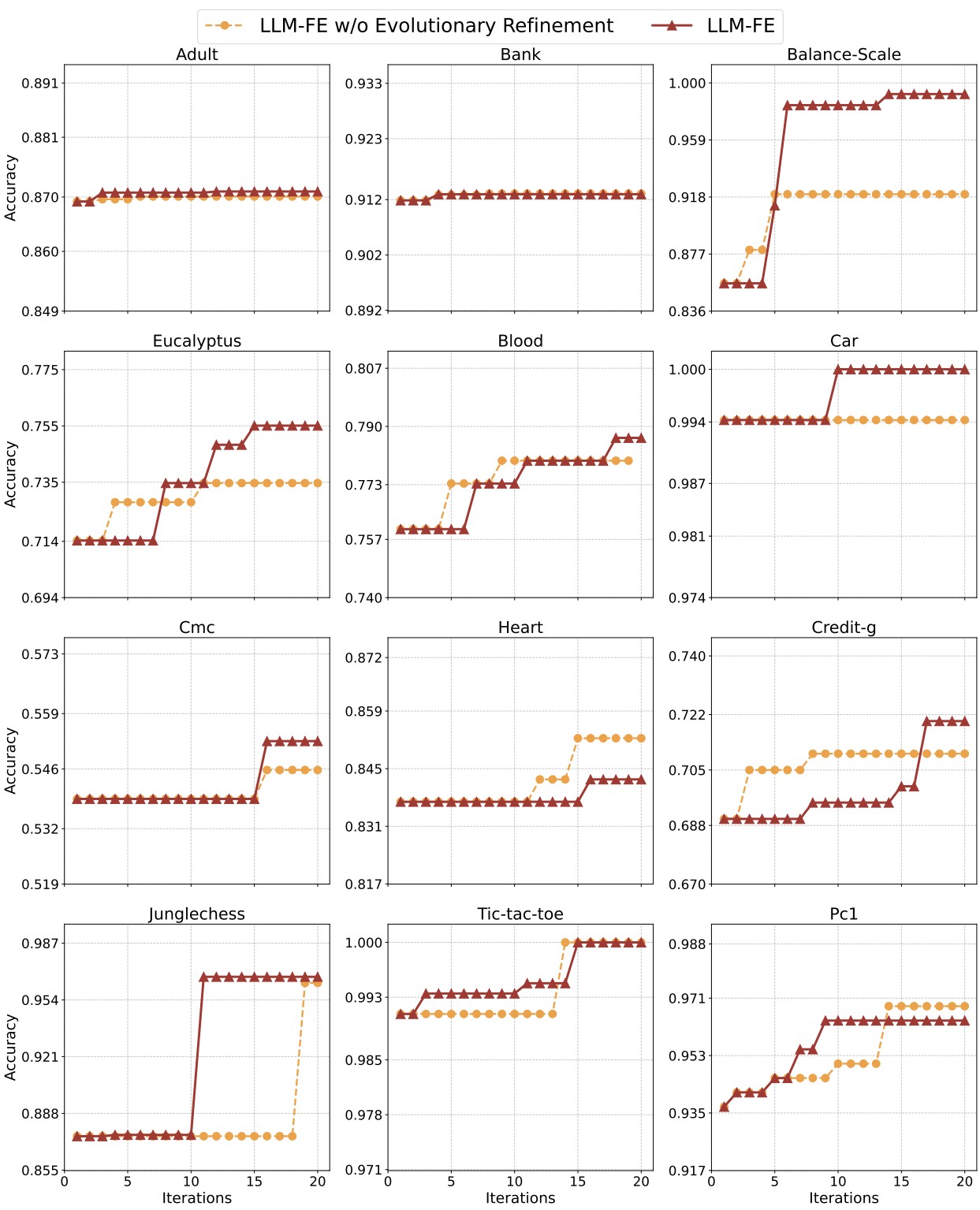

Figure 12: **Performance Trajectory Analysis.** Validation Accuracy progression for LLM-FE *w/o* evolutionary refinement and LLM-FE. LLM-FE demonstrates better validation accuracy, highlighting the advantage of evolutionary iterative refinement.

