# OpenReview forum: "LLM-FE: Automated Feature Engineering for Tabular Data with LLMs as Evolutionary Optimizers"
_TMLR — Accepted by TMLR_

### Review · Reviewer_cWFV · 2026-02-19

**Summary Of Contributions:**

This paper proposes LLM-FE, a framework that formulates automated feature engineering for tabular data as an LLM-guided evolutionary program search problem. The method iteratively generates feature transformation programs using an LLM, evaluates them via downstream model performance on a validation split, and maintains a multi-island memory buffer for evolutionary refinement. The approach integrates domain knowledge (via structured prompts), data-driven evaluation, and experience management.
Empirically, LLM-FE is evaluated on 16 classification and 10 regression datasets, compared against classical (AutoFeat, OpenFE) and LLM-based (CAAFE, FeatLLM, OCTree) baselines. Results show consistent improvements in classification and competitive gains in regression tasks. The paper further includes ablations, memorization analysis, operator bias analysis, efficiency evaluation, and cross-model/backbone generalization studies.
Key strengths include: (i) a clearly structured evolutionary framework with memory, (ii) broad empirical coverage across datasets and predictors, and (iii) dedicated analyses of memorization and operator bias.

**Audience:**

Yes

**Audience Explanation:**

The paper addresses automated feature engineering for tabular data, a widely relevant problem in applied machine learning and data-centric AI. Given the strong practical importance of tabular learning in industry and the increasing interest in LLM-driven optimization and evolutionary prompting, the proposed framework lies at the intersection of LLM reasoning, evolutionary search, and structured-data modeling.

Researchers interested in LLM-as-optimizer paradigms, automated data science, and hybrid symbolic–neural workflows would likely find the framework and analyses informative.

**Broader Impact Concerns:**

The paper includes an Impact Statement, which highlights benefits in reducing manual feature engineering effort and improving predictive performance. However, several ethical considerations merit further elaboration:

- Data Privacy Risks: The method includes serialized data samples in prompts. In real-world deployments, sending potentially sensitive tabular data to external LLM APIs may pose privacy and regulatory concerns. The paper does not provide operational guidance for privacy-preserving prompting.
- Automation and Human Oversight: By automating domain-informed feature discovery, the framework may reduce expert involvement. While efficiency gains are beneficial, the risks of spurious but high-performing transformations in safety-critical domains (e.g., healthcare datasets) are not discussed.

These concerns do not invalidate the contribution but suggest that the Broader Impact section could be expanded to include clearer deployment safeguards and ethical usage recommendations.

**Claims And Evidence:**

Yes

**Claims Explanation:**

The core claim that LLM-FE improves predictive performance over baselines is supported by empirical evidence on 16 classification datasets and 10 regression datasets. In classification, LLM-FE achieves the best mean rank (1.47) across datasets, and improvements over the base model are consistently reported. In regression, LLM-FE achieves the lowest mean rank among compared methods, although comparisons are restricted to classical feature engineering baselines due to missing regression implementations for LLM-based baselines.

The claim that evolutionary refinement contributes substantially is supported by ablation results, which show the largest performance drop when evolutionary refinement is removed, and by trajectory analysis, demonstrating continued improvement across iterations compared to early plateau without refinement.

Claims regarding memorization robustness are supported by experiments on datasets constructed to test memorization. The paper also supports claims of bias mitigation with operator frequency analysis.

However, statistical significance testing beyond mean ± std is not provided, and regression comparisons lack LLM-based baselines. Thus, while evidence is generally convincing, certain claims (e.g., broad “state-of-the-art” superiority across task types) are somewhat limited by evaluation scope.

**Requested Changes:**

**Clarify formalization and split protocol**

- Explicitly reconcile the bilevel formulation in Sec. 3.1 (Eqs. 1–2) with the evolutionary loop in Algorithm 1. Clearly define whether the optimization variable is the program $T$, and how $\pi_\theta$​ functions within the bilevel objective.
- Clarify clustering and sampling procedures in Sec. 3.4 (Boltzmann sampling, signature definition, temperature schedule).

**Strengthen regression comparisons**

- If feasible, implement or adapt LLM-based baselines (e.g., CAAFE-style methods) for regression tasks.
- Alternatively, explicitly scope claims in Abstract and Conclusion (Sec. 6) to clarify that regression comparisons are limited to classical methods (Sec. 4.4; Table 3).

**Add statistical significance analysis**

- Include paired statistical tests (e.g., Wilcoxon signed-rank across datasets) for Table 2 and Table 3 comparisons.
- Report confidence intervals or effect sizes in addition to mean ± std across splits.
- Highlight where improvements are statistically significant vs. marginal.

**Improve fairness and efficiency reporting**

- Report single-best-program results alongside ensemble results (Sec. 4.3) to disentangle ensemble effects from feature quality.
- Provide clearer runtime measurement details (hardware, inclusion/exclusion of LLM API latency) for Fig. 4.
- Clarify stopping/convergence criteria for LLM-based baselines beyond “20 samples” (Sec. 4.2).

**Expand broader impact and data privacy discussion**

- Discuss privacy implications of including serialized data samples in prompts

**Paper Writing**

- The authors misuse the commands \citet and \citep in several places. For example, at the end of the second line in the introduction, "particularly when working with tabular data Domingos (2012).", we should use \citep rather than \citet here. Similar issues are found in Table 1, when mentioning existing methods, and the last line of page 2 ("GPT-3.5-Turbo OpenAI (2023)"). I kindly request the authors to double-check all similar issues in their manuscript. Usually, we use \citet when the author is the subject/object of the sentence, and \citep for citations placed at the end of a sentence or clause.
- The appendices are not referenced in sequential order within the main text. For example, Appendix E is referenced before Appendix B and Appendix A. Academic convention requires appendices to be cited in the order they appear (A → B → C → …). The current ordering may confuse readers and suggests insufficient proofreading. I do recommend that the authors reorder appendix citations in the main text so they follow a strictly increasing alphabetical sequence.
- Several display equations are not followed by appropriate punctuation. In formal academic writing: **(1)** If a sentence continues after an equation, it should end with a comma. **(2)** If the equation completes the sentence, it should end with a period. Leaving equations without punctuation disrupts grammatical flow and is inconsistent with standard mathematical writing style.
- Many figures (especially multi-panel or operator-frequency plots) use font sizes that are noticeably smaller than the main body text. Usually, figure text should be clearly readable without zooming and should be roughly comparable in perceived size to the body text. Small fonts reduce accessibility and may impair reader comprehension.
- There are inconsistencies and potential inaccuracies in the reference list: **(1)** Some cited works that have already been published appear to still be referenced as arXiv preprints. For example, "Elephants never forget: Memorization and learning of tabular data in large language models." is accepted by COLM 2024, but the authors still cite its arXiv version in the current manuscript. **(2)** Within the same venue (e.g., NeurIPS), some entries include page numbers while others do not. For example, "Tom Brown, Benjamin Mann, Nick Ryder, Melanie Subbiah, Jared D Kaplan, Prafulla Dhariwal, Arvind Neelakantan, Pranav Shyam, Girish Sastry, Amanda Askell, et al. Language models are few-shot learners. Advances in neural information processing systems, 33:1877–1901, 2020." V.S. "Angelica Chen, David Dohan, and David So. Evoprompting: language models for code-level neural architecture search. Advances in Neural Information Processing Systems, 36, 2024." **(3)** Formatting conventions are not applied consistently across entries. This creates the impression of an unclean bibliography and weakens the professionalism of the submission. I suggest that the authors double-check and clean up the reference list.

---

> ### Author Response · Authors · 2026-03-16
> **Response to Reviewer cWFV (1/2)**
>
> We thank the reviewer for their time and thoughtful feedback, which helped us strengthen the manuscript.
>
> > Clarify formalization and split protocol.
>
> We thank the reviewer for this suggestion. We have modified Section 3.1 to clarify the connection between the bilevel formulation in Sec. 3.1 (Eqs. 1–2) and the evolutionary optimization procedure described in Algorithm 1, in our revised submission.
>
> > Clearly define whether the optimization variable is the program, and how functions within the bilevel objective. Clarify clustering and sampling procedures in Sec. 3.4 (Boltzmann sampling, signature definition, temperature schedule).
>
> We agree that these components would benefit from clearer exposition. Details about the sampling procedure are currently provided in Appendix B.2. To assist the readers, we have now updated Section 3.4 of the main paper to include the additional details mentioned by the reviewer.
>
> > Report single-best-program results alongside ensemble results to disentangle ensemble effects from feature quality.
>
> We want to clarify that **ensembling in LLM-FE is not an unfair advantage but a natural consequence of its multi-island design**. LLM-FE explores multiple independent search trajectories in parallel, each evolving toward different regions of the feature space. LLM-FE's islands are initialized differently and evolved independently, producing diverse and complementary feature sets that make the ensemble meaningful. This is fundamentally different from methods like CAAFE and OCTree, which follow a single optimization trajectory and produce one final feature set, making ensembling inapplicable to them by design. FeatLLM, on the other hand, also generates multiple programs produced independently, and thus, we take an ensemble for reporting FeatLLM results.
>
> > Provide clearer runtime measurement details
>
> We thank the reviewer for this clarification request. All runtime measurements in Section 5.1 were conducted on 4 NVIDIA RTX8000 GPUs. For LLM-based methods, the reported runtime covers the full pipeline from the start of the first iteration to the end of the last iteration, including LLM API latency, feature program execution, and model training and evaluation at each iteration. For classical methods, the reported runtime covers the complete feature generation, selection, and evaluation pipeline. We have added these details explicitly to Section 5.1 in the revised paper.
>
> > Clarify stopping/convergence criteria for LLM-based baselines beyond “20 samples” (Sec. 4.2).
>
> We thank the reviewer for this clarification request. All LLM-based methods are run for a fixed budget of 20 LLM samples with no early stopping or convergence criterion. This fixed-budget protocol was chosen deliberately to ensure a fair comparison across all methods, as it controls the total number of LLM queries regardless of each method's internal optimization strategy. The final reported performance is taken from the best-scoring program found within this budget. We have clarified this in Section 4.1 of the revised paper.
>
> > Expand broader impact and data privacy discussion. Discuss the privacy implications of including serialized data samples in prompts
>
> We thank the reviewer for raising this point. We agree that including serialized data samples can raise privacy concerns in sensitive domains. We note that this issue is not unique to our approach, as **prior LLM-based feature engineering methods**, such as CAAFE and FeatLLM, **also include data samples in their prompts**. Within LLM-FE, several mitigation strategies are possible; **LLM-FE can operate with locally deployed open-source LLMs**, which could eliminate external data transmission. Our experiments already show that LLM-FE performs well with open-source models. Second, **anonymized or synthetic representative samples can be used in prompts instead of raw data**. Finally, our ablation analysis (Section 4.5) shows that removing data samples from prompts leads to only a minor performance drop, indicating that **LLM-FE remains effective even when prompts exclude dataset examples entirely**. Moreover, data privacy in LLM-based systems is an ongoing research challenge and requires deeper investigation at both the methodological and system levels.
>
> > Automation and Human Oversight
>
> The automation of feature discovery is well-established in the AutoML literature, and LLM-FE continues this trajectory. LLM-FE crucially depends on user inputs such as task descriptions, feature metadata, and domain context, an example of human-in-the-loop AutoML. Rather than replacing domain experts, LLM-FE is designed to augment them, allowing data scientists to focus on higher-level problem-solving and decision-making. In safety-critical domains, LLM-FE's outputs should be treated as expert-reviewable hypotheses rather than directly deployable transformations. We have expanded the Impact Statement of our revision to include these points.

---

> ### Author Response · Authors · 2026-03-16
> **Response to Reviewer cWFV (2/2)**
>
> > Strengthen regression comparisons.
>
> We thank the reviewer for raising this. As mentioned in Section 4.4, we note that LLM-based baselines are not included for regression tasks due to the lack of available implementations. We would like to elaborate further on this limitation here. FeatLLM focuses on classification datasets, with no regression evaluations reported, as it generates features to infer class likelihood, predicting discrete class labels instead of continuous outputs. Similarly, CAAFE does not support regression in its released implementation. The repository (`data.py` file) explicitly raises an exception for regression datasets:
>
> ```
> if entry["NumberOfClasses"] == 0.0:
>  raise Exception("Regression not supported")
> ```
>
> We also note that the public OCTree repository does not provide regression datasets or runnable code for regression tasks. To address the reviewer’s concern as directly as possible, we implemented our own regression evaluation for OCTree and obtained the following results, which we will include in Table 3 (results reported in RMSE) in the revised paper:
>
> | Dataset | Base | OpenFE | OCTree | LLM-FE |
> |---|---|---|---|---|
> | airfoil_self_noise [10^0] | 1.572 ± 0.084 | 1.631 ± 0.111 | 1.572 ± 0.079 | **1.451 ± 0.059** |
> | bike [10^1] | 4.094 ± 0.096 | _4.089 ± 0.140_ | 4.094 ± 0.096 | **3.985 ± 0.084** |
> | cpu_small [10^0] | 2.857 ± 0.223 | 2.822 ± 0.190 | 2.832 ± 0.192 | **2.733 ± 0.249** |
> | crab [10^0] | 2.325 ± 0.094 | _2.221 ± 0.010_ | 2.280 ± 0.087 | **2.211 ± 0.124** |
> | diamond [10^2] | 5.479 ± 0.063 | _5.384 ± 0.084_ | 5.479 ± 0.063 | **5.356 ± 0.134** |
> | forest-fires [10^1] | 0.163 ± 0.009 | _0.161 ± 0.013_ | 0.162 ± 0.007 | **0.156 ± 0.008** |
> | housing [10^4] | 4.845 ± 0.191 | _4.628 ± 0.105_ | 4.845 ± 0.191 | **4.525 ± 0.260** |
> | insurance [10^3] | 5.269 ± 0.260 | 5.085 ± 0.286 | **4.969 ± 0.331** | _5.069 ± 0.392_ |
> | plasma_retinol [10^2] | _2.352 ± 0.196_ | 2.363 ± 0.195 | 2.362 ± 0.204 | **2.278 ± 0.248** |
> | wine [10^0] | 0.639 ± 0.006 | _0.631 ± 0.009_ | 0.639 ± 0.006 | **0.612 ± 0.007** |
>
> We have updated Table 3 in the revised manuscript to strengthen the regression evaluation.
>
> > Statistical significance testing
>
> We thank the reviewer for this suggestion; however, we would like to note that **none of the LLM-based methods (CAAFE, FeatLLM, OCTree) report any such statistical tests**, and so we believe that it is important but not done for this domain.  To address the reviewer's request, we conducted one-tailed Wilcoxon signed-rank tests across 25 paired observations per dataset, obtained from five random seeds across five-fold cross-validation. LLM-FE achieves statistically significant improvement over the base model on all 10 out of 10 regression datasets (p < 0.001) and 8 out of 15 classification datasets (p < 0.05), and 2 classification datasets (credit-g and communities) showing marginal improvement. We report 95% confidence intervals alongside mean ± std, with bold entries indicating statistically significant improvements. We have updated these results in the revision in Table 14 (Appendix D.4).
>
> | Dataset | Base | LLM-FE |
> |---|---|---|
> | adult | 0.872 ± 0.002 | **0.874 ± 0.003** |
> | bank | 0.906 ± 0.002 | **0.907 ± 0.003** |
> | breast-w | 0.955 ± 0.014 | **0.963 ± 0.012** |
> | blood | 0.747 ± 0.023 | 0.743 ± 0.024 |
> | car | 0.992 ± 0.005 | **0.998 ± 0.004** |
> | cdc-diabetes | 0.849 ± 0.001 | **0.849 ± 0.001** |
> | cmc | 0.527 ± 0.029 | 0.527 ± 0.024 |
> | communities | 0.699 ± 0.020 | _0.703 ± 0.019_ |
> | covtype | 0.871 ± 0.002 | **0.878 ± 0.001** |
> | credit-g | 0.754 ± 0.028 | _0.758 ± 0.027_ |
> | eucalyptus | 0.659 ± 0.029 | **0.671 ± 0.029** |
> | heart | 0.861 ± 0.022 | 0.857 ± 0.026 |
> | myocardial | 0.785 ± 0.025 | 0.788 ± 0.029 |
> | pc1 | 0.931 ± 0.010 | **0.937 ± 0.008** |
> | vehicle | 0.761 ± 0.028 | 0.767 ± 0.030 |
>
> | Dataset | Base | LLM-FE |
> |---|---|---|
> | forest-fires | 1.649 ± 0.116 | **1.567 ± 0.116** |
> | housing | 48014 ± 1184 | **44221 ± 1441** |
> | insurance | 5280 ± 306 | **5117 ± 353** |
> | bike | 40.782 ± 1.218 | **39.759 ± 1.373** |
> | wine | 0.637 ± 0.019 | **0.613 ± 0.019** |
> | crab | 2.309 ± 0.092 | **2.215 ± 0.097** |
> | diamond | 548.227 ± 10.368 | **536.522 ± 13.072** |
> | airfoil-self-noise | 1.547 ± 0.127 | **1.435 ± 0.113** |
> | cpu-small | 2.833 ± 0.210 | **2.718 ± 0.219** |
> | plasma-retinol | 233.565 ± 23.751 | **223.968 ± 26.756** |
>
> > * Fixing the usage of \citet and \citep
> > * Appendices referenced in sequential order
> > * Punctuation for Equations
> > * Fixing font sizes for figures
> > * Updating the arxiv preprints with official published venues
>
> We thank the reviewer for these corrections, and we have fixed these issues in the updated manuscript.

---

> > ### Comment · Reviewer_cWFV · 2026-04-05
> >
> > I thank the authors for their detailed responses. Most of my concerns are addressed properly.

---

### Review · Reviewer_ZXJy · 2026-02-20

**Summary Of Contributions:**

This paper introduces LLM-FE, an LLM-based automated feature engineering method for tabular data. The proposed method leverages LLMs to generate feature transformation programs, evaluates them using downstream model performance, and iteratively refines candidate features using an experience buffer and multi-population evolutionary search. Experiments show over classical and LLM-based baselines, with generalization across predictive models and LLM backbones.

This paper provides a comprehensive empirical evaluation across 26 tabular datasets, multiple predictive models, and different LLM backbones (GPT-3.5 and Llama-3.1). It also provides ablation and analysis studies that isolate the contributions of domain knowledge, evolutionary refinement, and data examples, while also examining memorization risks and operator bias.

However, the novelty is limited, as the proposed approach shares significant similarities with existing frameworks such as OCTree. Additionally, the execution time improvements are unclear, and the evaluation relies on relatively weak LLM backbones, raising questions about the full potential and generality of the approach.

**Audience:**

Yes

**Audience Explanation:**

The proposed framework demonstrates consistent performance improvements, and the ablation studies and analysis of domain knowledge, evolutionary refinement, and operator bias offer useful insights into components underlying effective LLM-guided feature discovery.

**Claims And Evidence:**

No

**Claims Explanation:**

The novelty of the proposed framework is limited relative to existing baselines such as OCTree, and the paper does not provide a precise methodological comparison that distinguishes LLM-FE. Also, the execution time improvement is significant, especially on the arrhythmia dataset. However, such improvement is not justified. Finally, the evaluation relies primarily on relatively weaker LLMs such as GPT-3.5 and Llama-3.1-8B, without demonstrating whether the proposed framework provides consistent gains with stronger modern models.

**Requested Changes:**

1. Methodological comparison between LLM-FE and existing works, especially OCTree. (Critical for acceptance)
2. Explain that the execution time improvement of LLM-FE is significant. (Important, would strengthen the work)
3. Experiments on SOTA LLMs. (Important, would strengthen the work)

---

> ### Author Response · Authors · 2026-03-16
> **Response to Reviewer ZXJy (1/2)**
>
> We thank the reviewer for their time and thoughtful feedback, which helped us strengthen the manuscript.
>
> > Methodological comparison.
>
> Thank you for raising this point. A methodological comparison with CAAFE and OCTree is included in Appendix A (earlier Appendix E), but we acknowledge it may not have been sufficiently emphasized in the main text. We will highlight it more clearly in the revised version, and appreciate the opportunity to clarify it more explicitly.
>
> - **Parallel Exploration:** CAAFE and OCTree refine a single candidate rule or feature iteratively. LLM-FE instead adopts a multi-island evolutionary strategy where **multiple candidate feature programs are evolved in parallel**, enabling *simultaneous exploration of diverse directions in the feature space* rather than a single refinement trajectory.
>
> - **Population-Based Memory:** LLM-FE maintains multiple evolving populations and a memory of high-performing feature programs. OCTree and CAAFE do not maintain a population of candidates; they refine one structure at a time without mutation and crossover.
>
> - **Feedback and Refinement Design:** LLM-FE explicitly applies **LLM-guided mutation and crossover over sampled high-performing individuals from different islands**. OCTree, in contrast, relies on a new decision tree that is trained at each iteration to suggest modifications to the current rule. This process remains centered on a single evolving candidate and does not incorporate evolutionary recombination mechanisms.
>
> - **Empirical Results:** The impact of this design is visible in Tables 2 and 3, where LLM-FE's multi-island strategy mitigates premature convergence by preserving diversity across islands, whereas OCTree's single-path refinement is more trajectory-dependent and prone to local optima, resulting in a higher mean rank across datasets.
>
> - **Feature Complexity:** Kuken et al. [1] state that LLMs exhibit a significant bias toward simple feature operators, thus impacting model performance. As shown in Table 1, LLM-FE produces complex features unlike OCTree and CAAFE. Specifically, 45% of LLM-FE's features qualify as complex under Küken et al.'s [1] definition, compared to < 5% for CAAFE and OCTree.
>
> > Execution time improvements.
>
> We thank the reviewer for raising this point. We want to clarify two distinct points: the efficiency of LLM-FE relative to baselines, and the specific behavior on the arrhythmia dataset. (i) **Pareto Efficiency**: The efficiency analysis in **Section 5.1 and Figure 3 shows that LLM-FE lies on the Pareto frontier of performance vs. runtime**. LLM-FE achieves higher predictive accuracy than OCTree at a similar runtime, and outperforms CAAFE and OpenFE primarily due to its multi-island search strategy for feature discovery. (ii) **Dataset-specific improvement**: On the arrhythmia dataset, we want to clarify that the ✗ symbols reflect two distinct failure modes: **for classical methods, ✗ indicates execution timeout** (>12 hours) attributed to the dataset's 279-dimensional feature space, causing a combinatorial explosion. For **LLM-based baselines** (CAAFE, FeatLLM, OCTree), **✗ it reflects runtime execution errors**, not timeout, as these methods failed to run to completion on this dataset due to the large feature space. Thus, the improvement cannot be solely attributed to execution time. Additionally, CAAFE and OCTree generate only one feature discovery program for every LLM call, whereas LLM-FE samples 3 outputs for every LLM call, thereby reducing the number of LLM calls by one-third, thereby reducing the overall cost. We have updated the text to clarify this in the table caption for Table 2 and updated Section 5.1 to include more details

---

> ### Author Response · Authors · 2026-03-16
> **Response to Reviewer ZXJy (2/2)**
>
> > Experiments on SOTA LLMs.
>
> Thank you for the valuable suggestion. We demonstrate the generalizability of our approach by *evaluating it across diverse downstream prediction models, including tree-based models, neural networks, and transformer-based architectures, as well as across different LLM backbones*. Our method improves performance across variations, highlighting LLM-FE's generalizability. Furthermore, we evaluated LLM-FE with three more recent LLMs: GPT-4o-mini, Gemini-2.5-Flash, and Qwen-2.5-72B-Instruct. The results are shown in the table below:
>
> | Dataset | XGBoost| GPT-4o-mini | Qwen2.5-72b | Gemini-2.5-Flash |
> | - | - | - | - | - |
> | breast-w | 0.956 ± 0.012 | **0.974 ± 0.006** | **0.969 ± 0.011** | **0.961 ± 0.011**    |
> | blood | 0.742 ± 0.012 | **0.750 ± 0.029** | **0.749 ± 0.022** | **0.747 ± 0.023**    |
> | car | 0.995 ± 0.003 | **1.000 ± 0.000** | **0.999 ± 0.001** | **0.997 ± 0.005**    |
> | cmc  | 0.528 ± 0.029 | **0.534 ± 0.016** | **0.538 ± 0.016** | **0.534 ± 0.024**    |
> | credit-g  | 0.751 ± 0.019 | **0.775 ± 0.022** | **0.764 ± 0.028** | **0.755 ± 0.012**    |
> | eucalyptus | 0.655 ± 0.024 | **0.678 ± 0.028** | **0.670 ± 0.022** | **0.659 ± 0.032**    |
> | heart  | 0.858 ± 0.013 | **0.863 ± 0.023** | 0.857 ± 0.016 | 0.847 ± 0.012    |
> | pc1  | 0.931 ± 0.004 | **0.934 ± 0.004** | **0.935 ± 0.008** | **0.934 ± 0.009**    |
> | vehicle  | 0.754 ± 0.016 | **0.770 ± 0.020** | **0.761 ± 0.027** | **0.766 ± 0.026**    |
>
> | Dataset | XGBoost| GPT-4o-mini | Qwen2.5-72b | Gemini-2.5-Flash |
> |------------------------|-------------------|--------------------|--------------------|--------------------|
> | bike [10^1] | 4.094 ± 0.096 | **3.928 ± 0.186** | **4.027 ± 0.322** | **3.960 ± 0119** |
> | crab [10^0] | 2.325 ± 0.094 | **2.193 ± 0.132** | **2.262 ± 0.203** | **2.273 ± 0.154** |
> | housing [10^4] | 4.845 ± 0.191 | **4.430 ± 0.126** | **4.813 ± 0.344** | **4.769 ± 0.393** |
> | insurance [10^3] | 5.269 ± 0.260 | **5.100 ± 0.351** | **5.108 ± 0.296** | 5.351 ± 0.776 |
> | wine [10^0] | 0.639 ± 0.006 | **0.616 ± 0.006** | **0.610 ± 0.004** | **0.6330 ± 0.005** |
>
> LLM-FE improves over the base model across both regression and classification datasets, similar to the observations with GPT-3.5-Turbo and Llama-3.1-8B, strengthening LLM-FE's generalizability across LLM backbones, including more recent models. We include this table in the revised paper in Appendix D.2 (Table 11).
>
> ---
>
> **References**
>
> [1] Küken et al. "Large language models engineer too many simple features for tabular data.", Arxiv 2024

---

### Review · Reviewer_ypYJ · 2026-03-06

**Summary Of Contributions:**

The paper considers the problem of feature engineering for tabular data in a prediction task. Traditional methods (i.e., AutoFE) are limited since they rely on fixed, manual design features, combining search space, and fail to incorporate the domain knowledge (as designed by humans). Recent approach using LLMs to incorporate domain-knowledge into this process, but relies on direct prompting or single-path refinement, which leads to effectively leveraging memory or iterative data-driven feedback.

To solve this, the authors proposed LLM-FE, a framework that uses LLMs for feature engineering with interactive feedback and evolutionary strategy. Their method works as follows: (1) Hypothesis generation: an LLM reacts as a "mutator", to generate feature transformations (written as a Python codes), (2) data-driven evaluation: the python codes are executed to augment the training data, and a downstream predictive model (i.e., XGBoost) is trained and evaluated on the validation split to yield a score, and (3) the algorithm maintains an external memory buffer divided into independent sub-populations (they call this __islands__). High-performing transformations (in the Python code form) are saved into islands and subsequently sampled to serve as in-context samples for future LLM prompts, guiding the model toward iterative refinement.

The authors then test their method on 26 datasets (including both classification and regression tasks), and compare against non-LLM methods (AutoFeat, OpenFE) as well as LLM methods (CAAFE, FeatLLM, OCTree). Besides, the authors also provide a couple of ablation studies.

**Additional Comments:**

Again, since it is TMLR, I put little weight on the novelty of this methodology. I also won't expect that this method is superior in any use case or any dataset. However, I expect to see a more complete, rigorous evaluation of this method, in all aspects (the good/bad/ugly). Right now, I feel like the experiments are not enough, and I hope to see more in the rebuttal phase. Also, feel free to correct me if I am missing something.

**Audience:**

Yes

**Audience Explanation:**

Yes, I think feature engineering is the most critical and challenging task for tabular data, which is widely used by many traditional fields.

**Claims And Evidence:**

Yes

**Claims Explanation:**

The claims are generally supported by empirical evidence. Some of them are not that persuasive (which will be discussed below), but generally, they are acceptable.

**Requested Changes:**

I will go over some of my concerns and then request changes based on those concerns accordingly.

### Main concerns
1. Methodology design: as far as I understand, the methodology is like this: at step 0, we partition the feature set into __islands__. In step $t$, we pick an island, and we do feature generation using a subset of features in this island. The authors said that this method could prevent local minima (somehow). However, if I understand correctly, this method has a fatal weakness, since if there are two features on two different islands that could make a good combination, they would never be combined. Can the authors comment on these points? If this is correct, then the experimental design has to be curated massively to address this issue.

2. The point above also raises another point: the initial partition has a major impact on the features generated and the results in each run, and I also expect it to be a bit volatile. Therefore, the results have to be evaluated more carefully (i.e., more runs or statistical power evaluation). When I look at the results, I see that the gain is minimal compared to the methods, and with current runs, it is hard to conclude that this method is actually better.

3. Currently, the authors only use two (slightly outdated) LLM models for evaluation. Besides, they noted that those models are biased towards simple feature transformation (like adding, subtracting, etc). Though funding could be an issue, I want to see some results on frontier models to see if: (1) this bias is still a case, or (2) the frontier models tend to __over-complicate__ the feature engineering (i.e., generating too complicated features). Maybe try with some lite models (like Gemini Flash 3) to save the cost.

4. On hyperparameter optimization (HPO): I see that the authors do conduct HPO, but only for the base model and their own LLM-FE model. However, they do not do HPO for the baselines (i.e., AutoFeat, OpenFE, ...) This is another big concern for me. Can the authors comment on this point?
.

### Minor issues

1. Citations: Here are some format linter errors, which are good to get fixed:

   1.1. Typos: There are some typos for citations (i.e., at the end of page 2).
   1.2. Bibliography: Some of the bibliography uses the non-official version. For example, “Nam et al.  Optimized feature generation for tabular data via llms with decision tree reasoning, 2024” uses the arxiv version, while it is in NeurIPS'24.

2. Technical issues: In Eq. 2, the authors should write $f^* \in \\arg \\min_{f} \\mathcal{L}_{f}  (f (\\mathcal{T}(\\mathcal{X}_t), \\mathcal{Y}))$ instead, since the argmin is a set, and we do not know if this set is unique. I know that this one is one of the few mathematical notations in this draft, but it is good to write it correctly to improve the presentation.

---

> ### Author Response · Authors · 2026-03-16
> **Response to Reviewer ypYJ (1/2)**
>
> We thank the reviewer for their time and thoughtful feedback, which helped us strengthen the manuscript.
>
> > Methodology design
>
> We thank the reviewer for this question. The design of dividing a population into subpopulations to improve solution quality and maintain search diversity has been studied previously in the field of evolutionary computation. LLM-FE adopts a multi-island framework and builds on works like FunSearch [1] and LLM-SR [2], which use a similar isolated island architecture to prevent premature convergence and improve model performance. The islands in LLM-FE are organized to **structure the search space during optimization and to guide exploration in the discovery process**. Table 16 (Appendix E.4) confirms this empirically as the performance is stable across k = 1, 3, and 5 islands, with k = 3 providing the best exploration–exploitation balance. We would like to clarify that, in our method, the *islands do not partition the feature set*. **Each island has access to the full original feature set**; what differs across islands is the search trajectory, not the available variables. Therefore, a combination of any two original features can, in principle, be discovered within any island, including features that might appear to belong to different regions of the search space. A minor tweak like cross-island migration of top-performing candidates could potentially provide improvements. Nevertheless, the **results in Tables 2 and 3 already show performance improvements across datasets and tasks with the current design**, indicating that the architecture is effective as-is. Moreover, in Table 16, Appendix E.4, we conducted
>
> > More careful evaluation of results
>
> As clarified above, the islands do not partition the feature set, and every island has access to all original features and is initialized identically (Section 3.4, Appendix B.1). Beyond Tables 2 and 3, the paper provides extensive evidence that LLM-FE's gains are consistent and not limited to a particular run configuration. **Section 5.2, Table 4 shows improvements across XGBoost, MLP, and TabPFN, under GPT-3.5-Turbo and Llama-3.1-8B**. Additionally, **Appendix D.2, Tables 10-12 extend to CatBoost and Logistic Regression**; we also **added newer LLMs such as GPT-4o-mini, Gemini-2.5-Flash, and Qwen2.5-72B-Instruct (Table 11)**. These results highlight that LLM-FE's gains are not volatile and generalizable across experiment settings. **Appendix E.2 provides an interpretability analysis showing that LLM-FE's generated features impact the final prediction** performance of the model using SHAP value analysis. We also conducted statistical significance testing across datasets in Tables 2 and 3. To evaluate whether LLM-FE produces statistically significant improvements over the base model, we conduct one-tailed Wilcoxon signed-rank tests across 25 paired observations per dataset, obtained from five random seeds across five-fold cross-validation. LLM-FE achieves statistically significant improvement over the base model on all 10 out of 10 regression datasets (p < 0.001) and 8 out of 15 classification datasets (p < 0.05), and 2 classification datasets (credit-g and communities) showing marginal improvement. We report 95% confidence intervals alongside mean ± std, with bold entries indicating statistically significant improvements. We have updated these results in the revision in Table 14 (Appendix D.4).
>
> | Dataset | Base | LLM-FE |
> |---|---|---|
> | adult | 0.872 ± 0.002 | **0.874 ± 0.003** |
> | bank | 0.906 ± 0.002 | **0.907 ± 0.003** |
> | breast-w | 0.955 ± 0.014 | **0.963 ± 0.012** |
> | blood | 0.747 ± 0.023 | 0.743 ± 0.024 |
> | car | 0.992 ± 0.005 | **0.998 ± 0.004** |
> | cdc-diabetes | 0.849 ± 0.001 | **0.849 ± 0.001** |
> | cmc | 0.527 ± 0.029 | 0.527 ± 0.024 |
> | communities | 0.699 ± 0.020 | _0.703 ± 0.019_ |
> | covtype | 0.871 ± 0.002 | **0.878 ± 0.001** |
> | credit-g | 0.754 ± 0.028 | _0.758 ± 0.027_ |
> | eucalyptus | 0.659 ± 0.029 | **0.671 ± 0.029** |
> | heart | 0.861 ± 0.022 | 0.857 ± 0.026 |
> | myocardial | 0.785 ± 0.025 | 0.788 ± 0.029 |
> | pc1 | 0.931 ± 0.010 | **0.937 ± 0.008** |
> | vehicle | 0.761 ± 0.028 | 0.767 ± 0.030 |
>
> | Dataset | Base | LLM-FE |
> |---|---|---|
> | forest-fires | 1.649 ± 0.116 | **1.567 ± 0.116** |
> | housing | 48014 ± 1184 | **44221 ± 1441** |
> | insurance | 5280 ± 306 | **5117 ± 353** |
> | bike | 40.782 ± 1.218 | **39.759 ± 1.373** |
> | wine | 0.637 ± 0.019 | **0.613 ± 0.019** |
> | crab | 2.309 ± 0.092 | **2.215 ± 0.097** |
> | diamond | 548.227 ± 10.368 | **536.522 ± 13.072** |
> | airfoil-self-noise | 1.547 ± 0.127 | **1.435 ± 0.113** |
> | cpu-small | 2.833 ± 0.210 | **2.718 ± 0.219** |
> | plasma-retinol | 233.565 ± 23.751 | **223.968 ± 26.756** |

---

> ### Author Response · Authors · 2026-03-16
> **Response to Reviewer ypYJ (2/2)**
>
> > Bias Towards Simple Features using Frontier Models
>
> We thank the reviewer for this suggestion. We want to clarify that Küken et al. [3], which we cite in the paper, measures operator frequency across tabular datasets for two proprietary models (GPT-4o-mini and Gemini-1.5-flash) and two open-source models (Llama3.1-8B and Mistral-7B). Their findings reveal that simplicity bias persists more in frontier models like GPT-4o-mini and Gemini-1.5-Flash, compared to the smaller open-sourced models. In response to the reviewer's comments, we further conducted additional experiments using three newer models: GPT-4o-mini, Gemini-2.5-Flash, and Qwen2.5-72B to understand the behavior, with results shown below:
>
> | Operator | Qwen2.5-72B | LLM-FE | Gemini-2.5-Flash | LLM-FE | GPT-4o-mini | LLM-FE |
> |----------|---------|--------|--------------|--------|--------------|--------|
> | abs      | 0.04 | 0.03 | 0.05 | 0.04 | 0.03 | 0.02 |
> | add      | 0.09 | 0.13 | 0.18 | 0.12 | 0.01 | 0.16 |
> | divide   | **0.51** | 0.32 | **0.43** | 0.24 | **0.52** | 0.41 |
> | multiply | **0.31** | 0.32 | **0.23** | 0.12 | **0.24** | 0.24 |
> | subtract | 0.05 | 0.06 | 0.11 | 0.06 | 0.10 | 0.10 |
> | sq/sqrt   | 0.000 | 0.015 | 0.000 | 0.089 | 0.012 | 0.03 |
> | residual | 0.00 | 0.00 | 0.00 | 0.00 | 0.00 | 0.00 |
> | sigmoid  | 0.00 | 0.01 | 0.00 | 0.03 | 0.00 | 0.00 |
> | groupbythenmax | 0.00 | 0.01 | 0.00 | 0.00 | 0.00 | 0.00 |
> | log | 0.00 | 0.02 | 0.00 | 0.14 | 0.000 | 0.03 |
> | groupby thenmin | 0.00 | 0.00 | 0.00 | 0.00 | 0.00 | 0.01 |
> | min/max | 0.00 | 0.01 | 0.01 | 0.06 | 0.00 | 0.01 |
> | groupby thenmean | 0.00 | 0.08 | 0.00 | 0.09 | 0.00 | 0.01 |
>
> As seen from the results, divide and multiply alone account for ~65–80% of the generated feature operators, and complex operators appear at near-zero frequency, confirming that simplicity bias persists regardless of model scale. While simple operators also remain present in LLM-FE's results, LLM-FE actively steers all three frontier models toward complex operators that are rarely seen in their base versions, with no evidence of over-complication. Furthermore, *downstream performance results for three frontier models are reported in our response to Reviewer ZXJy.*
>
> > Hyperparameter optimization (HPO)
>
> We thank the reviewer for raising this, and we conducted HPO for OpenFE and CAAFE using the same Optuna-based search procedure described in Appendix C.1, with identical hyperparameter spaces. Results for XGBoost are shown below across the five classification datasets used in our HPO analysis:
>
> | Dataset            |  Base  |  OpenFE | CAAFE | LLM-FE |
> |--------------------|---------------|--------------- |---------------|---------------|
> | eucalyptus         | 0.681 ± 0.029 | 0.687 ± 0.017  | 0.679 ± 0.017 | 0.678 ± 0.020 |
> | credit-g           | 0.746 ± 0.023 | 0.754 ± 0.019  | 0.755 ± 0.011 | 0.784 ± 0.017 |
> | cmc                | 0.552 ± 0.030 | 0.551 ± 0.013  | 0.543 ± 0.017 | 0.578 ± 0.021 |
> | blood-transfusion  | 0.790 ± 0.010 | 0.777 ± 0.016  | 0.789 ± 0.021 | 0.805 ± 0.009 |
> | vehicle            | 0.760 ± 0.016 | 0.810 ± 0.016  | 0.770 ± 0.021 | 0.801 ± 0.033 |
>
> Given the limited rebuttal period, we were able to conduct the HPO robustness analysis for XGBoost only. We will extend this analysis to MLP and include it in Appendix D.1 in the final camera-ready version.
>
> > Minor issues - Citations
>
> We thank the reviewer for raising this. We have corrected the arXiv preprint versions with their official published venues in the final camera-ready version, and fixed the use of  \citet and \citep in the paper.
>
> > Technical issues: In Eq. 2
>
> We thank the reviewer for this correction, and we have updated the camera-ready version accordingly.
>
> ----
>
> **References**
>
> [1] Romera-Paredes et al.  "Mathematical discoveries from program search with large language models", Nature 2024
>
> [2] Shojaee et al. "LLM-SR: Scientific Equation Discovery via Programming with Large Language Models", ICLR 2025
>
> [3] Küken et al. "Large language models engineer too many simple features for tabular data.", Arxiv 2024

---

### Decision · Action_Editor_u845 · 2026-04-06

**Recommendation:** Accept as is

**Additional Comments:**

The reviewers find that the claims of this work are generally supported by empirical evidence, the empirical evaluation is comprehensive, and the proposed framework demonstrates consistent performance improvements. However, they also raise concerns regarding novelty, methodology design, methodological comparison with existing work (OCTree), evaluation settings and details, results on frontier and state-of-the-art LLM models, better regression comparisons, hyperparameter optimization, statistical significance testing, clarification and writing, as well as broader impact and data privacy discussion.

The authors have provided a thorough response with new results to address these concerns. After the rebuttal, all reviewers voted to lean toward accepting this work.

I have read this paper in detail and agree with all reviewers that this work is a valuable contribution to TMLR, and especially all its main claims are well supported by extensive experiments. Therefore, I recommend accepting this work as is. For the camera-ready version, please carefully incorporate all the discussion and analysis into the paper, and open-source the code to facilitate reproducibility of the proposed method.

**Audience:**

Yes

**Audience Explanation:**

All reviewers believe some individuals in TMLR's audience could be interested in the findings of this paper.

**Claims And Evidence:**

Yes

**Claims Explanation:**

This work proposes LLM-FE, a large language model (LLM) based automated feature engineering approach for tabular data. The key idea is to formulate feature engineering as a program search problem and combine LLM-based program generation with evolutionary optimizers to iteratively search for impactful features. Experimental results show that the proposed LLM-FE approach consistently outperforms state-of-the-art methods on feature engineering across diverse tabular learning tasks.

All reviewers believe the claims made in this paper are supported by convincing and clear evidence. Please refer to the comment below for more details.